# A single-cell atlas of the miracidium larva of *Schistosoma mansoni* reveals cell types, developmental pathways, and tissue architecture

**Teresa Attenborough[1,2†], Kate A Rawlinson[1,3*†], Carmen L Diaz Soria[1], Kirsty Ambridge[1], Geetha Sankaranarayanan[1], Jennie Graham[1], James A Cotton[1,4], Stephen R Doyle[1], Gabriel Rinaldi[1,5], Matthew Berriman[1,2*]**

[1]Wellcome Sanger Institute, Wellcome Genome Campus, Hinxton, United Kingdom; [2]School of Infection and Immunity, College of Medical, Veterinary & Life Sciences, University of Glasgow, Glasgow, United Kingdom; [3]Josephine Bay Paul Center, Marine Biological Laboratory, Woods Hole, United States; [4]School of Biodiversity, One Health and Veterinary Medicine, College of Medical, Veterinary & Life Sciences, University of Glasgow, Glasgow, United Kingdom; [5]Department of Life Sciences, Aberystwyth University, Aberystwyth, United Kingdom

**\*For correspondence:**
krawlinson@mbl.edu (KAR);
Matt.Berriman@glasgow.ac.uk
(MB)

†These authors contributed
equally to this work

**Competing interest:** The authors
declare that no competing
interests exist.

**Reviewing Editor:** Utpal
Banerjee, University of California,
Los Angeles, United States

**Abstract** *Schistosoma mansoni* is a parasitic flatworm that causes the major neglected tropical disease schistosomiasis. The miracidium is the first larval stage of the life cycle. It swims and infects a freshwater snail, transforms into a mother sporocyst, where its stem cells generate daughter sporocysts that give rise to human-infective cercariae larvae. To understand the miracidium at cellular and molecular levels, we created a whole-body atlas of its ~365 cells. Single-cell RNA sequencing identified 19 transcriptionally distinct cell clusters. In situ hybridisation of tissue-specific genes revealed that 93% of the cells in the larva are somatic (57% neural, 19% muscle, 13% epidermal or tegument, 2% parenchyma, and 2% protonephridia) and 7% are stem. Whereas neurons represent the most diverse somatic cell types, trajectory analysis of the two main stem cell populations indicates that one of them is the origin of the tegument lineage and the other likely contains pluripotent cells. Furthermore, unlike the somatic cells, each of these stem populations shows sex-biased transcriptional signatures suggesting a cell-type-specific gene dosage compensation for sex chromosome-linked loci. The miracidium represents a simple developmental stage with which to gain a fundamental understanding of the molecular biology and spatial architecture of schistosome cells.

## eLife assessment

This is a **valuable** study in which the authors provide an expression profile of the human blood fluke, Schistosoma mansoni. A strength of this **solid** study is in its inclusion of in situ hybridisation to validate the predictions of the transcript analysis.

## Introduction

The parasitic blood fluke *Schistosoma mansoni* (Phylum Platyhelminthes) is a causative agent of the neglected tropical disease schistosomiasis, which is responsible for a chronic disease burden equivalent to 1.0–2.6 million Disability Adjusted Life Years and over 11,000 human deaths annually (***WHO, 2023***; ***IHME, 2019***). The life cycle of *S. mansoni* is complex, involving parasitic stages in two hosts (a

snail and a mammal), with short-lived, freshwater infectious larval stages in between. Mature male and female worms reside in the portal veins of the mammalian host, and fertilisation of the zygote occurs inside the female worm (*Jurberg et al., 2009*). Embryogenesis proceeds as the egg is laid in the host and continues over the ensuing 6 days (*Ashton et al., 2001*). During this time, the egg passes through the host vascular endothelium into the intestine and is passed in faeces into the outside environment (*Costain et al., 2018*). If the egg lands in freshwater, a miracidium larva hatches out, swims and infects a snail host. Once inside the snail, the miracidium transforms into the mother sporocyst, and its stem cells (historically known as 'germinal cells') divide and differentiate to produce many daughter sporocysts that migrate within the snail. In the daughter sporocysts, a second round of embryogenesis generates the cercariae larvae by asexual reproduction from a stem cell. The cercariae emerge from the snail, swim towards and infect water-exposed mammals including humans, penetrating the skin. At the point of infection, the cercariae shed their tails, and once inside the mammal the tegument (skin) is remodelled during metamorphosis to the schistosomulum stage. The schistosomulum then moves into the blood vessels and, over the following 5 weeks, grows into a juvenile stage that migrates from the lungs to the liver and to the hepatic portal vein where sexually dimorphic male and female worms pair, mature, and mate. Thus, during the development from egg to adult, the body axes are established three times, once during the development of the miracidia, once during the development of the daughter sporocyst and finally during the development of the cercariae. Additionally, there are two rounds of embryogenesis, first from fertilised zygote to the mature miracidium, and second, the neo-embryogenesis from a single stem cell in the daughter sporocyst to a mature cercaria.

Enabled by the advent of single-cell RNA sequencing (scRNA-seq), molecular classification of cell types has been initiated for *S. mansoni* (*Wang et al., 2018*; *Tarashansky et al., 2019*; *Li et al., 2021*; *Diaz Soria et al., 2020*; *Wendt et al., 2020*; *Diaz Soria et al., 2024*). These studies have opened the door to investigate schistosome developmental cell biology and to identify new targets for controlling infection and transmission of the disease. To date, scRNA-seq studies have identified heterogeneity amongst stem cells that drive schistosome development and reproduction, classified a variety of differentiated cell types across the life cycle and revealed several key genes and their native cell types that are essential for parasite survival and propagation (*Nanes Sarfati et al., 2021*). Single-cell RNA-seq studies for the whole animal have only been carried out on intra-molluscan and intra-mammalian stages to date: the mother sporocyst (*Diaz Soria et al., 2024*), the schistosomulum (*Diaz Soria et al., 2020*), juvenile (3.5 weeks post-infection of mouse, *Li et al., 2021*), and adult (*Wendt et al., 2020*). Single-cell studies on stem cells specifically, however, have only been undertaken in the mother sporocyst (*Wang et al., 2018*) and juvenile stages (2.5 and 3.5 weeks post-infection)(*Tarashansky et al., 2019*).

An obvious gap in the coverage of the life cycle is the lack of single-cell data for the earliest developmental time point currently accessible – the miracidium larva; earlier embryonic stages are inaccessible due to the impenetrable egg capsule. The miracidium is a short-lived, non-feeding stage that relies on internal stores of energy to sustain swimming, and host location and infection (*Maldonado and Acosta-Matienzo, 1948*; *Chernin, 1968*; *Pan, 1980*). The role of the miracidium and its differentiated somatic cells is to carry stem cells to a favourable environment inside the snail where they can proliferate to generate daughter sporocysts, and subsequently cercariae. If the miracidium fails to infect a snail within 12 hours it dies.

At the ultrastructural level, a rich set of cell types have been previously identified in the miracidium and classified into eight major systems: musculature, nervous, penetration gland cells, excretory, interstitial (parenchyma), stem/germinal cells, epithelial (ciliary plates and epidermal ridges of the tegument cells), and terebratorium (an apical modified epidermal plate with sensory structures and gland openings) (*Pan, 1980*).

To characterise this critical transmission stage, we have generated a single-cell transcriptome atlas for the miracidium. Our atlas provides molecular definitions for all described cell types (*Pan, 1980*) and indicates the functions of different cell types within tissues. Multiplexed in situ validation of cell-type marker genes enabled the cell types to be spatially resolved within the larva. Furthermore, the nuclear segmentation of the cells expressing tissue-specific genes has revealed the relative contribution of each tissue type to the animal and each cell in the larva has been assigned to a tissue. A key finding is the identification of two stem cell populations in a miracidium, and advances in our understanding of the genome (*Buddenborg et al., 2021*) has enabled us to determine that these two

stem populations are transcriptionally distinct between male and female larvae due to sex-linked gene expression. To predict the fate of these two stem populations we carried out a trajectory analysis, combining scRNA-seq data from miracidia and mother sporocysts, and this indicated that one stem cell population likely contains the pluripotent cells that will develop into the cercariae, while the other population gives rise to the somatic tissues, including the tegument. Similarities in gene expression with the adult suggest that there is a tegument developmental programme that is redeployed at multiple stages of the complex life cycle.

## Results

### Single-cell RNA-seq of 20,478 miracidia cells reveals 19 cell types

There are ~365 nuclei in a miracidium (median = 365, larva #1: 342, larva #2: 365, larva #3: 377). Some cell types, such as the apical gland and epithelial/tegument cells, are multinucleate; and others, such as the ciliary plates may be anucleate (*Pan, 1980*, but cf *Meuleman et al., 1978*). Therefore, the number of nuclei provides an approximation for the actual number of cells (*Figure 1A*).

We performed scRNA-seq on a pool of ~20,000 mixed-sex miracidia, collected within 4 hr of hatching from eggs (*Figure 1B*). Following dissociation, four samples of the cell suspensions were collected; two were enriched for live cells using fluorescence-activated cell sorting, and two were left unsorted to capture the maximum cellular diversity for the atlas (*Figure 1B*). Using the droplet-based 10X Genomics Chromium platform, transcriptome-sequencing data were generated from a total of 33,391 cells, of which 20,478 passed strict quality control (QC) filters, resulting in a median of 858 genes and 1761 median unique molecular identifier (UMI) counts per cell (*Supplementary file 1a*). Given that an individual miracidium comprises ~365 cells (*Figure 1A*), the number of quality-controlled cells theoretically represents >56-fold coverage of each cell in the larva.

Using Seurat (version 4.3.0) (*Hao et al., 2021*), 19 distinct clusters of cells were identified, along with putative marker genes best able to discriminate between the populations (*Figure 1C, D* and *Supplementary file 1b, c*). We used Seurat's JackStraw and ElbowPlot, along with molecular cross-validation to select the number of principal components (PCs), and Seurat's clustree to select a resolution where clusters were stable (*Hao et al., 2021*). Manually curated lists of previously defined cell-type-specific genes from later developmental stages (*Diaz Soria et al., 2020*; *Wendt et al., 2020*; *Nanes Sarfati et al., 2021*) were compared against the list of putative markers generated in this analysis. The expression of genes from both these lists in the miracidia single-cell data was then examined to identify the cell types that each Seurat cluster represented. Based on these markers, the 19 clusters were resolved into the following cell populations: two muscle-like (3934 cells), five resembling neurons (3255 cells), one ciliary plate (40 cells), one tegumental (366 cells), one protonephridial (539 cells), two parenchymal (1698 cells), and seven clusters resembling stem cells (10,686 cells) (*Figure 1C*).

There were contributions from sorted and unsorted samples in almost all clusters (except ciliary plates). We found that some cell/tissue types had similar recovery from both methods (e.g. Stem A, Muscle 2, and Tegument), others were preferentially recovered by sorting (e.g. Neuron 1, Neuron 4, and Stem E), and some were depleted by sorting (e.g. Parenchyma 1, Protonephridia, and Ciliary plates) (*Figure 1—figure supplement 1*, *Supplementary file 1d*). This variation in recovery, therefore, enabled us to maximise the discovery and inclusion of different cell types in the atlas.

### Orthogonal body wall muscle fibres are transcriptionally distinct

The two muscle clusters were identified based on the expression of previously described muscle-specific genes; *paramyosin* (Smp_085540), *troponin t* (Smp_179810), and *titin* (Smp_126240) (*Wendt et al., 2020*; *Diaz Soria et al., 2020*; *Diaz Soria et al., 2024*), as well as other differentially expressed markers; for example, a putative collagen alpha chain (Smp_159600) (*Figure 2A*). A crp1/csrp1/crip1 homologue (Smp_087250), encoding a conserved transcriptional regulator of muscle (*Tarashansky et al., 2021*) was also expressed in both muscle clusters. High expression of *paramyosin* (*PRM*; Smp_085540) was seen in muscle cell clusters 1 and 2 (*Figure 2A*). In situ hybridisation (ISH) (*Figure 2B, C*; *Figure 2—video 1*) and image analysis (*Figure 2—video 2*) identified 74 nuclei surrounded by *PRM* transcripts (i.e. ~19% of cells in a miracidium are muscle cells). Counterstaining with phalloidin (to label F-actin filaments) revealed an orthogonal grid of circular and longitudinal body wall muscles (BWMs) and from this we identified which *PRM*+ nuclei belonged to each muscle

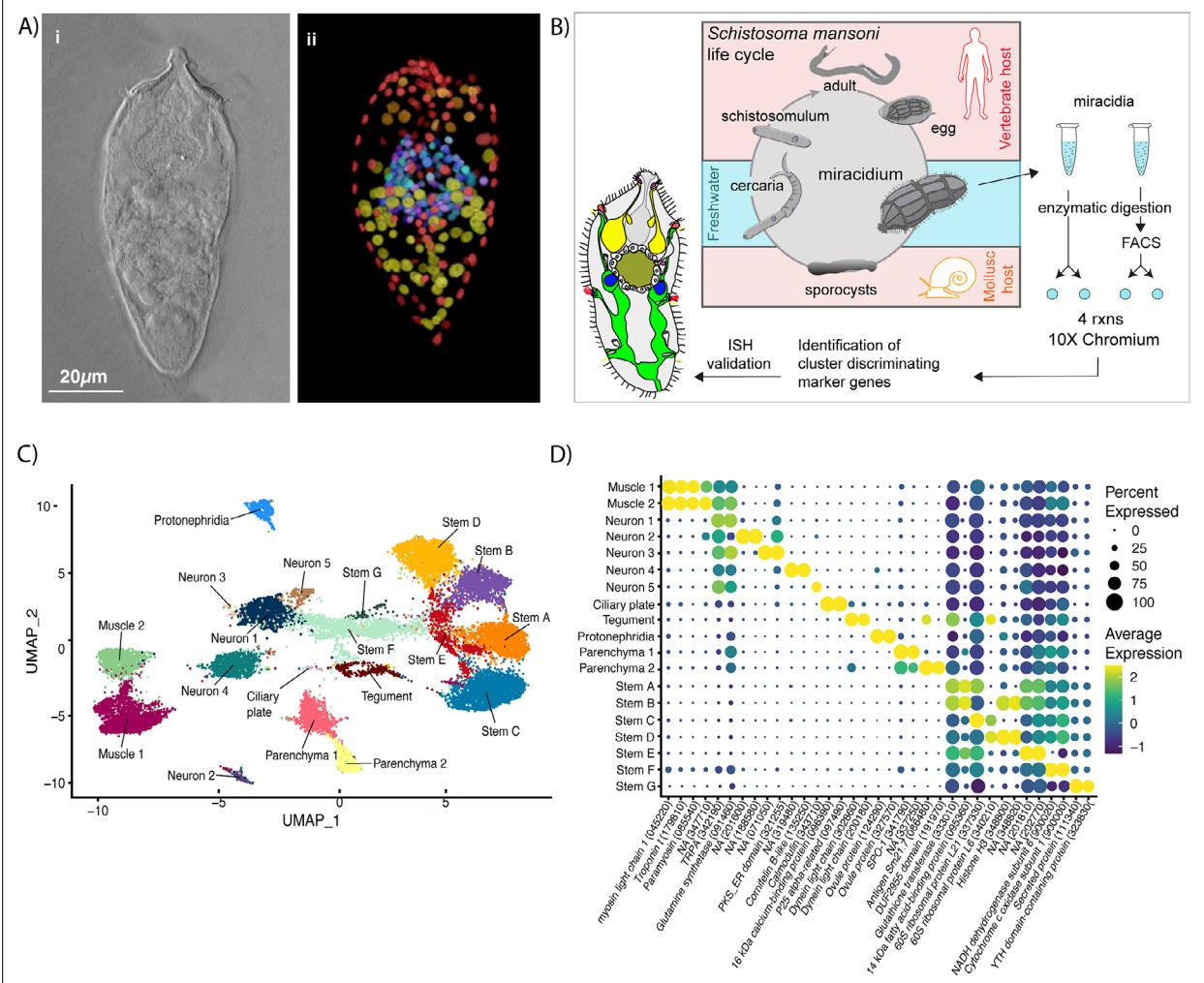

**Figure 1.** Identification of 19 transcriptionally distinct cell types in the miracidium. (**A**) The miracidium is composed of ~365 cells; (**i**) differential interference contrast (DIC) microscopy image of miracidium, (**ii**) a maximum intensity projection (MIP) of a confocal z-stack of miracidium, stained with 4',6-diamidino-2-phenylindole (DAPI), and with nuclei segmented to enable counting (larval anterior pole at the top in all images). (**B**) Experimental scheme describing the parasite's life cycle (images of developmental stages not to scale), parasite dissociation, single-cell analysis of miracidium, and validation pipeline. An average of 9975 miracidia per sample were dissociated; two samples were enriched for live cells (propidium iodide negative) using fluorescence-activated cell sorting (FACS), another two samples were unsorted. Cells were loaded according to the 10X Chromium single-cell 3' protocol. Clustering was carried out to identify distinct populations and population-specific markers. Validation of population-specific markers was performed by in situ hybridisation (ISH). (**C**) Uniform Manifold Approximation and Projection (UMAP) representation of 20,478 miracidium single cells. Cell clusters are coloured and distinctively labelled by cluster identity. (**D**) Gene expression profiles of the top population markers identified for each cell cluster (gene identifiers shown in parenthesis but with 'Smp_' prefix removed for brevity). The colours represent the expression level from yellow (high expression) to dark blue (low expression). Gene expression has been log-normalised and scaled using Seurat(v. 4.3.0). The sizes of the circles represent the percentages of cells in those clusters that expressed a specific gene.

The online version of this article includes the following figure supplement(s) for figure 1:

**Figure supplement 1.** Contributions of sorted and unsorted cells to Seurat clusters.

subtype. The nuclei of the circular muscles formed two distinct bilaterally symmetrical lines that ran peripherally from pole to pole of the larva (~28 nuclei in total). Thirty-three more *PRM*⁺ nuclei sat regularly spaced between the circular muscle nuclei and corresponded to longitudinal muscles. Another 13 *PRM*⁺ nuclei formed a unilateral cluster adjacent to the apical gland cell (identifiable by its four nuclei; *Pan, 1980*; *Figure 2Cii*; *Figure 2—videos 1 and 2*).

Muscle cluster 1 (2407 cells) was distinguished from muscle cluster 2 by markedly higher expression of a Kunitz-type protease inhibitor (Smp_052230; *Figure 2A*), and ISH showed Smp_052230 transcripts along the two peripheral lines of circular muscle nuclei (*Figure 2D*). Subpopulations of muscle

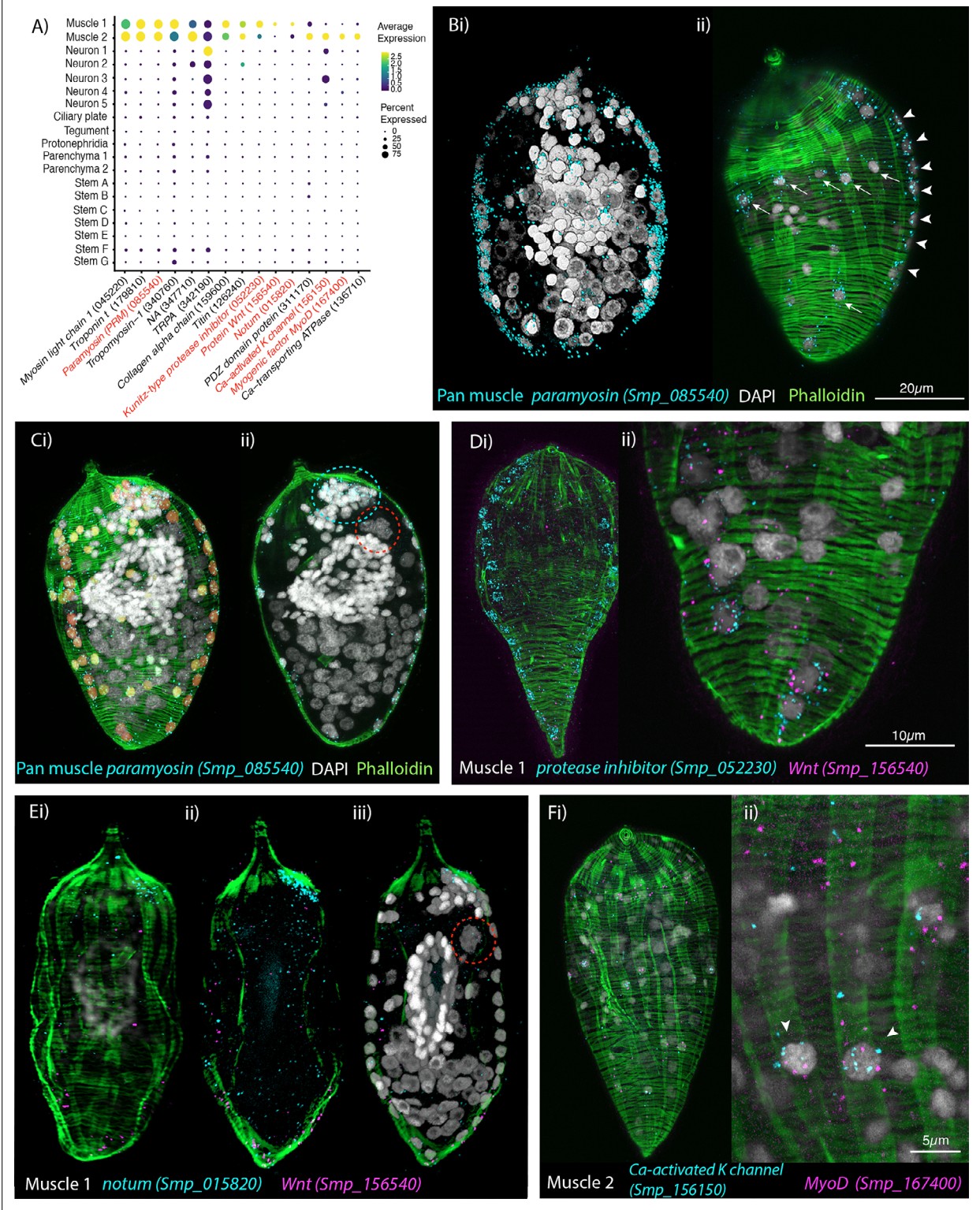

**Figure 2.** Orthogonal body wall muscles (BWMs) are transcriptionally distinct. (**A**) Dot plot depicting the expression profiles of specific or enriched marker genes in the two muscle clusters. Genes validated by in situ hybridisation (ISH) are marked in red. Gene identifiers shown in parenthesis but with 'Smp_' prefix removed for brevity. Gene expression has been log-normalised and scaled using Seurat(v. 4.3.0). (**B, i, ii**) Wholemount ISH (WISH) of paramyosin *PRM*+ and counterstaining with phalloidin reveals the nuclei of the circular BWMs, which form two distinct bilaterally symmetrical lines that run peripherally from the pole of the larva to the other (arrowheads in ii). The longitudinal BWM nuclei (ii – arrows) are spaced regularly between the lines of the circular BWM nuclei and their actin fibres run orthogonally to the circular muscles. These patterns were seen in 100% of individuals

*Figure 2 continued on next page*

*Figure 2 continued*

examined, *n* = 50. (**C, i**) Segmentation of the *PRM*⁺ cells in one miracidium shows 74 muscle cells in total: ~28 circular (segmented in orange), 33 longitudinal (yellow), and (**ii**) 13 in an anterior unilateral cluster (inside dashed cyan line) adjacent to the apical gland cell (identifiable by its four nuclei, inside dashed red circle). (**D**) Expression of markers for muscle 1 cluster; (**i**) *Kunitz-type protease inhibitor* is expressed in circular BWM and *wnt-11-1* is expressed in seven *Kunitz-type protease inhibitor*⁺ cells at the posterior pole; (**ii**) close-up of posterior end of another miracidium showing the co-expression of *Kunitz-type protease inhibitor* and *wnt-11-1*. In 100% of individuals examined, *n* = 30. (**E, i–iii**) *Notum*, an inhibitor of *wnt* signalling, is expressed highly at the opposite pole to *wnt-11-1* in the cluster of 13 muscle nuclei adjacent to the apical gland cell (nuclei in dashed red circle). In 100% of individuals examined, *n* = 30. (**F**) Expression of markers for muscle 2 cluster; a *calcium-activated potassium channel* (Smp_156150) and the transcription factor *myoD* (Smp_167400) show co-expression in longitudinal BWMs (arrowheads). In 100% of individuals examined, *n* = 30. Scale shown in B also applies to C, Di, E, and Fi.

The online version of this article includes the following video(s) for figure 2:

**Figure 2—video 1.** Confocal z-stack of wholemount in situ hybridisation (WISH) of pan-muscle marker *paramyosin* (*PRM* Smp_085540) (cyan), counterstained with phalloidin (green), and DAPI (white), reveals which *PRM*⁺ nuclei belong to the circular and longitudinal body wall muscles.
https://elifesciences.org/articles/95628/figures#fig2video1

**Figure 2—video 2.** Segmentation of *paramyosin* (Smp_085540) (cyan) positive cells; the nuclei of the circular body wall muscles are highlighted in orange, the nuclei of the longitudinal body wall muscles are highlighted in yellow.
https://elifesciences.org/articles/95628/figures#fig2video2

**Figure 2—video 3.** Confocal z-stack of wholemount in situ hybridisation (WISH) of muscle cluster 1 markers *Kunitz-type protease inhibitor* (Smp_052230) (cyan) and *Wnt-11-1* (Smp_156540) (magenta), counterstained with phalloidin (green), and DAPI (white) shows expression in circular body wall muscles.
https://elifesciences.org/articles/95628/figures#fig2video3

**Figure 2—video 4.** Maximum intensity projection from confocal z-stack of wholemount in situ hybridisation (WISH) of muscle cluster 1 markers *notum* (Smp_015820) (cyan) and *Wnt-11-1* (Smp_156540) (magenta) counterstained with phalloidin (green) showing expression at opposite poles.
https://elifesciences.org/articles/95628/figures#fig2video4

**Figure 2—video 5.** Confocal z-stack of wholemount in situ hybridisation (WISH) of muscle cluster 2 markers *calcium-activated potassium channel* (Smp_156150) (cyan) and *myoD* (Smp_167400) (magenta), counterstained with phalloidin (green) and DAPI (white) shows expression in longitudinal body wall muscles.
https://elifesciences.org/articles/95628/figures#fig2video5

cluster 1 also expressed orthologues of *Wnt-11-1* (Smp_156540) and *Notum* (Smp_015820), known anterior–posterior body axis patterning genes, where Notum acts as an inhibitor of Wnt signalling (*Petersen and Reddien, 2009*; *Kakugawa et al., 2015*; *Figure 2A*). *Notum* (Smp_015820) was highly expressed in the cluster of 13 muscle nuclei adjacent to the apical gland cell (*Figure 2E*). *Wnt-11-1* (Smp_156540) was expressed at the posterior pole, in seven of the Kunitz-type protease inhibitor-expressing circular muscle cells (*Figure 2D*; *Figure 2—video 3*). *Notum* was also expressed, albeit weakly, in some nuclei of circular muscles, including those expressing *Wnt-11-1* (*Figure 2—video 4*).

Muscle cluster 2 (1355 cells) was distinguished by higher expression of genes encoding a PDZ domain protein (Smp_311170) and *Calcium-transporting ATPase* (Smp_136710), as well as the unique expression of a putative calcium-activated potassium channel (Smp_156150), and the transcription factor *MyoD* (Smp_167400)(*Figure 2A*); ISH of the latter two genes showed expression only in the longitudinal BWMs (*Figure 2F*; *Figure 2—video 5*). Orthologues of other axial patterning genes, *Netrin receptor UNC5* (Smp_172470), *Wnt5* (Smp_145140), and a putative wnt inhibitor frizzled, frzb2 (Smp_062560) (*Witchley et al., 2013*), showed distinct expression in Muscle 2 cells.

The higher expression of Muscle 1 markers in circular BWMs and Muscle 2 genes in the longitudinal BWM revealed distinct transcriptomic signatures for these two types of myocyte that make up the orthogonal body wall musculature. In fact, differential expression analysis showed that *Wnt-11-1* (circular) and *MyoD* (longitudinal) were amongst the most differentially expressed genes between the two muscle clusters (*Supplementary file 1e*).

## Neural abundance and diversity

We uncovered five clusters expressing the neural markers *Complexin* (*cpx*; Smp_050220), *prohormone convertase 2* (*PC2*; Smp_077980), and Smp_068500 (a neural marker in adult worms, *Wendt et al., 2020*; *Figure 3A*). *Complexin* was expressed in and around at least 209 nuclei, indicating that 57% of cells in the miracidium were neurons; out of which 129 *cpx*⁺ nuclei formed the nuclear rind of the brain (or neural mass/ring), and the remaining 80 were situated peripherally, either anterior or

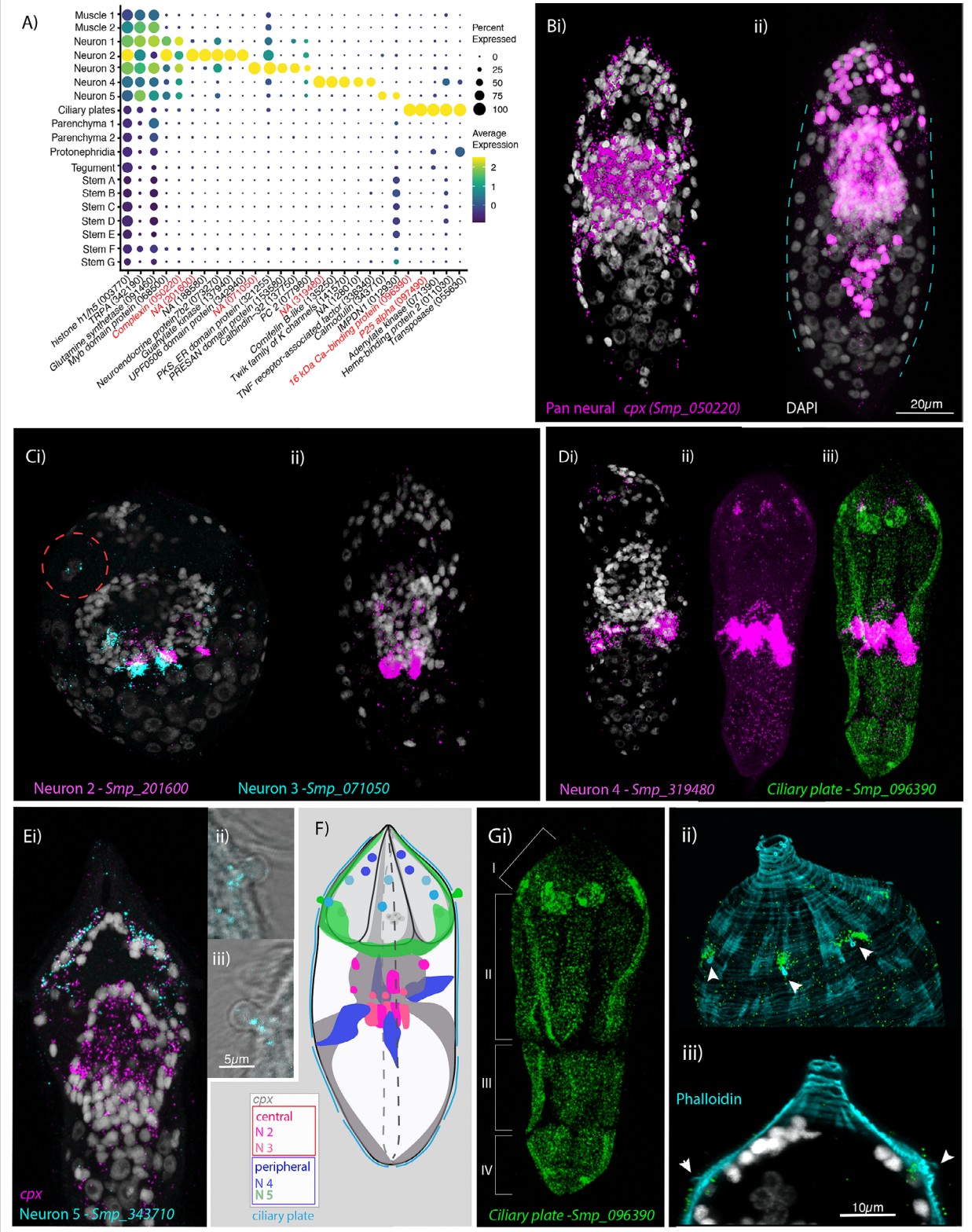

**Figure 3.** Neural complexity and ciliary plates in a simple larva. (**A**) Expression profiles of cell marker genes that are specific or enriched in the five neuronal clusters and ciliary plates. Genes validated by in situ hybridisation (ISH) are marked in red. Gene identifiers shown in parenthesis but with 'Smp_' prefix removed for brevity. Gene expression has been log-normalised and scaled using Seurat(v. 4.3.0). *Complexin* (*cpx*) is expressed in all five nerve clusters and is a neural marker in the adult (***Wendt et al., 2020***). (**B, i**) A dorso-ventral view of wholemount in situ hybridisation (WISH) of

*Figure 3 continued on next page*

*Figure 3 continued*

*cpx* shows expression in the brain, to the anterior – with projections around the three gland cells, and in two posterior clusters (100% of individuals examined, *n* = 50). (**ii**) Segmentation of *cpx*[+] cells (in magenta) in one miracidium (lateral view) reveals that at least 209 of the 365 cells that make up the larva are neural and that the clusters of *cpx* cells that are located posterior to the brain, and send projections to the posterior pole, sit perpendicular to the circular muscle cell nuclei (blue dashed lines). (**C**) Multiplexed ISH for top markers for neuron 2 (NPP 43, Smp_201600) and neuron 3 (NPP 26, Smp_071050) shows expression in multiple cells of the brain, including cells adjacent to each other. NPP 26 (Smp_071050) is also expressed in the apical gland cell (red dashed circle). 100% of individuals examined, *n* = 30. (**D**) The top marker for neuron 4, Smp_319480, is expressed in four cells anteriorly (arrows) and ~22 that sit lateral and just posterior to the brain; these later cells send projections into the brain and out to the body wall at the latitude between the second and third tiers of ciliary plates (**iii**) seen here expressing the ciliary plate marker gene Smp_096390. 100% of individuals examined, *n* > 30. (**E**) Neuron 5 marker Smp_343710, an EF-hand domain-containing protein, is expressed in (**i**) 10–20 cells whose nuclei sit outside of, and anterior to, the brain, in and around the paired lateral glands and their secretory ducts, and (**Eii, iii, F**) in a pair of bilaterally symmetrical bulbous protrusions. 100% of individuals examined, *n* = 30. (**F**) Summary schematic of the in situ expression of marker genes for the neural cell clusters. (**G, i**) A top marker for ciliary plates, a calcium-binding protein (Smp_096390), shows transcripts expressed in all the ciliary plates of all four tiers (brackets) and in six cells towards the anterior pole. (**ii, iii**) Counterstaining with phalloidin shows that the nuclei of these anterior cells sit beneath the body wall muscle and send a projection externally between the first and second tiers of ciliary plates (arrowheads). 100% of individuals examined, *n* > 30. Scale shown in B also applies to C, D, Ei, and Gi.

The online version of this article includes the following video and figure supplement(s) for figure 3:

**Figure supplement 1.** Subclustering the Neuron 1 cluster using self-assembling manifold (SAM) revealed multiple distinct subpopulations.

**Figure supplement 2.** Neurons.

**Figure supplement 3.** Phylogeny of metazoan opsins.

**Figure supplement 4.** STRING analysis of neural clusters and ciliary plates.

**Figure supplement 5.** The expression of two polycystin genes and a Sox transcription factor in Neuron 4 cells.

**Figure supplement 6.** Gene expression by transcripts per million in the ciliary plates and gland cells.

**Figure 3—video 1.** Confocal z-stack of wholemount in situ hybridisation (WISH) of markers for neuron cluster 2 (Smp_201600) (magenta) and neuron 3 (Smp_071050)(cyan), counterstained with DAPI (white) shows expression of both genes in cells whose nuclei form part of the nuclear rind of the brain.
https://elifesciences.org/articles/95628/figures#fig3video1

**Figure 3—video 2.** Confocal z-stack of wholemount in situ hybridisation (WISH) of ciliary plate marker *16 kDa calcium-binding protein* (Smp_096390), counterstained with phalloidin (green) and DAPI (white) shows expression in the ciliary plates and six previously undescribed cells that we term submuscular cells.
https://elifesciences.org/articles/95628/figures#fig3video2

**Figure 3—video 3.** Maximum intensity projection of confocal z-stack of wholemount in situ hybridisation (WISH) of ciliary plate marker *16 kDa calcium-binding protein* (Smp_096390), counterstained with phalloidin (green) and DAPI (white) shows expression in ciliary plates and submuscular cells.
https://elifesciences.org/articles/95628/figures#fig3video3

**Figure 3—video 4.** Maximum intensity projection from confocal z-stack of wholemount in situ hybridisation (WISH) of neuron 2 marker Smp_319480 (magenta), counterstained with DAPI (white), showing expression in clusters of cells posterior to the brain that extend into the brain and out to the periphery.
https://elifesciences.org/articles/95628/figures#fig3video4

**Figure 3—video 5.** Maximum intensity projection from confocal z-stack of wholemount in situ hybridisation (WISH) of neuron 4 marker Smp_319480 (magenta) and ciliary plate marker *16 kDa calcium-binding protein* (Smp_096390) (green), counterstained with DAPI (white), showing that the clusters of neuron 4 cells reach the periphery at the latitude between the second and third tiers of ciliary plates.
https://elifesciences.org/articles/95628/figures#fig3video5

**Figure 3—video 6.** Maximum intensity projection from confocal z-stack of wholemount in situ hybridisation (WISH) of neuron 5 marker Smp_319480 (cyan), counterstained with DAPI (white), showing expression in cells anterior to the brain and in and around the paired lateral glands and their secretory ducts.
https://elifesciences.org/articles/95628/figures#fig3video6

posterior to the brain (*Figure 3B*). Sensory organelles were identified for peripheral neural cell types. We were able to validate neuron clusters 2–5 using ISH of specific marker genes and performed an in silico analysis of Neuron 1.

Neuron cluster 1 was large (1470 cells), and expressed neuron-marker *complexin*, the neuroendocrine precursor *7B2* (Smp_073270) and an uncharacterised gene (Smp_068500). At this resolution, there were no clear marker genes restricted to Neuron 1. We explored whether Neuron 1 could be further subdivided into transcriptionally distinct cells by subclustering (*Figure 3—figure supplement 1*; *Supplementary file 1f*) using the self-assembling manifold (SAM) algorithm (*Tarashansky et al., 2019*) with ScanPy (*Wolf et al., 2018*), given its reported strength in discerning subtle variation in

gene expression (*Tarashansky et al., 2019*), although a similar topology was subsequently found using Seurat. Three subclusters of Neuron 1 expressed neuropeptide precursor genes amongst their top 5 markers; *PWamide* (Smp_340860) (*Wang et al., 2016*) in subcluster 14, *Neuropeptide F* (Smp_088360) in subcluster 8, and *Neuropeptide Y* (Smp_159950) in subcluster 13 (*Figure 3—figure supplement 1*); and each was expressed in the nuclear rind of the brain (*Figure 3—figure supplement 2*). Another subcluster was composed of putative photoreceptor cells (subcluster 7, *Figure 3—figure supplement 1*, *Supplementary file 1f*) because their top 10 markers include two opsins (Smp_180350 and Smp_332250) – identified here as a rhabdomeric opsin and a peropsin, respectively (*Figure 3—figure supplement 3*). Subcluster 7 also expressed other conserved phototransduction cascade genes, such as a 1-phosphatidylinositol 4,5-bisphosphate phosphodiesterase (Smp_245900) and a beta-arrestin (Smp_152280) (*Figure 3—figure supplement 1*, *Supplementary file 1f*).

Neuron 2 expressed two further rhabdomeric opsins (Smp_180030 and Smp_104210; *Figure 3—figure supplement 3*), but across the cluster of 282 cells, the most expressed marker was Smp_201600, identified as the neuropeptide precursor NPP-43 (*Koziol et al., 2016*). ISH revealed three strongly Smp_201600$^+$ cells, with nuclei in a median posterior location of the brain, and three cells with weaker expression in a more anterior position (*Figure 3C*, *Figure 3—video 1*). Other top markers include *7B2*, a guanylate kinase (Smp_137940) and a putative homeobox protein (Smp_126560).

Neuron 3 was a small cluster (47 cells) expressing *complexin* and *7B2*. The neuropeptide precursor NPP-26 (Smp_071050) (*Koziol et al., 2016*) was unique to and highly expressed throughout this population (*Figure 3A*). ISH showed expression in ~6 cells with projections that extended into the brain and to the periphery, three of which sat adjacent to three Neuron 2 cells. There was also weak expression in the four nuclei of the apical gland cell (*Figure 3Ci*, *Figure 3—video 1*). STRING analysis of the top 100 markers of Neuron 3 predicted two protein interaction networks with functional enrichment; these were associated with 'synapses' and 'calcium' (*Figure 3—figure supplement 4*). In Neuron clusters 1, 2, and 3, there was an enrichment for genes annotated with gene ontology (GO) terms related to 'neuropeptide signalling pathway' and 'neurotransmitter transport' (*Supplementary file 1g*). There were, therefore, at least five, and likely several more, transcriptionally distinct cell types in the brain mass (we validated three subclusters of Neuron 1, as well as Neurons 2 and 3).

Cells from neuron clusters 4 and 5 had nuclei peripheral to the brain. Neuron 4 formed a large (1140 cells), discrete cluster. The top marker was an uncharacterised gene, Smp_319480 (*Figure 3A*), that, based on protein-structure prediction using I-TASSER (*Yang and Zhang, 2015*), was similar to human mTORC1 (protein database [PDB] entry 5H64; TM-score 0.85). In situ expression revealed ~26 Smp_319480$^+$ cells; four at the anterior end, with projections leading to the body wall in the first tier of ciliary plates, and ~22 sat lateral and directly posterior to the brain. These latter cells formed four clusters of ~3–7 cells, with Smp_319480 transcripts distributed around the nuclei and along projections leading into the brain and extending out to the body wall terminating between the second and third tiers of ciliary plates (*Figure 3D*, *Figure 3—videos 4 and 5*). Twenty-three multi-ciliated nerve endings formed a girdle around the larva between the second and third tiers of ciliary plates (*Figure 3—figure supplement 2C*), and Smp_319480 transcripts were expressed at the base of these cilia (*Figure 3—figure supplement 2D*); some transcripts also reached the nuclei of circular BWMs (*Figure 3—figure supplement 2E*), suggesting these cells may function as sensory-motor neurons. Two genes, identified as top markers and co-expressed in many Neuron 4 cells, encode likely orthologues of *Polycystin 1* (Smp_165230) and *Polycystin 2* (Smp_165660) (*Figure 3—figure supplement 5*), with the latter prediction strongly supported by a Foldseek search (*van Kempen et al., 2024*) against the PDB. Polycystins 1 and 2 form a heterodimeric ion channel required for oscillating calcium concentrations within vertebrate cilia (*Kim et al., 2016*). A Sox transcription factor, Smp_301790, was also expressed in Neuron 4 and co-expressed with *Polycystin 1* and *2* (*Figure 3—figure supplement 5B*). Other top markers included a putative potassium channel (Smp_141570), a cAMP-dependent protein kinase (Smp_079010), a GTP-binding protein (Smp_045990), and an orthologue of human *fibrocystin-L* (Smp_303980). The GO terms 'cilium', 'regulation of apoptotic process', and 'potassium ion transmembrane transport' were significantly enriched, and STRING analysis showed networks enriched in 'plasma membrane-bound cell projection' (*Supplementary file 1g*, *Figure 3—figure supplement 4*).

Neuron 5 cluster comprised 276 cells. An EF-hand domain-containing protein, Smp_343710, was highly expressed in 55% of these cells. ISH showed expression in 10–20 cells whose nuclei sat outside of, and anterior to, the brain, in and around the paired lateral glands and their secretory ducts

(*Figure 3—video 6*). There was also expression in a pair of bilaterally symmetrical bulbous protrusions (called lateral sensory papilla and thought to be depth sensors, *Pan, 1980*; *Figure 3F*). STRING analysis revealed networks associated with 'synapses', 'ion channel binding', and 'cation transmembrane transporter activity' (*Figure 3—figure supplement 4*).

We expected to find a discrete cluster(s) for the penetration glands, and that it would show similarities to the neural clusters (as glandular cells arise from neuroglandular precursor cells in other animals, such as the sea anemone, *Nematostella vectensis*, *Steger et al., 2022*). There are three gland cells per miracidium – one apical and two lateral (*Pan, 1980*). While the top marker gene for Neuron 3 was expressed in the four nuclei of the apical gland (*Figure 3Ci*), and there was expression of a Neuron 5 marker in the lateral glands (*Figure 3Fi*), these genes were expressed in many other cells as well. This suggests that gland cells are either subclusters of Neurons 3 and 5 or they were not captured in the 10X GEMs. To characterise the transcriptomes of the gland cells, we carried out plate-based single-cell RNA-seq (New England BioLabs) on 37 manually selected gland cells (*Supplementary file 1h*). The biological process GO terms enriched in the most abundant genes (top 200 genes by median transcripts per million [TPM], *Supplementary file 1h*) included translation, protein folding and ATP synthesis coupled electron transport (*Supplementary file 1i*). The gland cells showed co-expression of neural markers (*Complexin* and *7B2*) and muscle markers *Myosin light chain 1* (Smp_045200), *Troponin t* (Smp_179810), and *Actin-2* (Smp_307020) (*Figure 3—figure supplement 6*). To identify genes that were specific to gland cells, we searched for differentially expressed genes in the gland cells compared to the other cells in the plate-based scRNA-seq data (i.e. ciliary plates and unknown cells) (*Supplementary file 1j and k*), and then looked at their expression in the 10X-based scRNA-seq data. A lack of, or minimal, expression of these markers in the 10X-based scRNA-seq data, would support the hypothesis that they were specific to the gland cells. Five venom antigen-like (VAL) genes (belonging to the SCP/TAPS or (CAP) superfamily) were expressed in the gland cells: Smp_120670 (VAL 5) and Smp_070250 (VAL 15) in over 60% of the gland cells, and Smp_176180 (VAL 9), Smp_002630 (VAL 2), and Smp_331830 (SCP protein) in less than 50%. Many gland cells expressed multiple VAL genes (*Figure 3—figure supplement 6*). In the 10X-based scRNA-seq data, very few cells expressed these VAL genes (one to five cells for each gene, distributed between Neurons 1 and 5, Muscle 1, Parenchyma 2, and Stems F and G clusters). The VAL genes expressed in the gland cells are uniquely expressed during the egg, miracidia, and sporocyst stages (*Yoshino et al., 2014*; *Lu and Berriman, 2018*; *Buddenborg et al., 2023*), and it has been reported that VAL 9 (Smp_176180) induces the differential expression of an extracellular matrix remodelling gene in the snail host (*Yoshino et al., 2014*). Other differentially expressed genes in the gland cells included Smp_303400 (an HSP70 homologue), Smp_179250 (*α-galactosidase/α-N-acetylgalactominidase*), Smp_317530 (uncharacterised protein), and two putative major egg antigens Smp_185680 and Smp_303690 (*Figure 3—figure supplement 6*). Eight putative secreted proteins previously identified in miracidia and in vitro transformed miracidia (*Wang et al., 2016*; *Wu et al., 2009*) were expressed in the glands cells (and not in other cell types based on the 10X data), and five of these showed relatively high expression; Smp_302170 (*HSP70* homologue), Smp_303400 (*HSP70* homologue), Smp_049230 (*HSP20*), Smp_304250 (*purine nucleoside phosphorylase*), and Smp_302180 (putative heat shock protein 70)(*Figure 3—figure supplement 6*). Typical of venom glands in other animals, the transcriptomes of the miracidia gland cells are enriched in genes involved in protein translation, stabilisation, and stress response to cope with mass protein production (*Zancolli et al., 2022*).

## Ciliary plate gene marker identifies six submuscular cell bodies

A small cluster of 40 cells (almost exclusively from the unsorted samples) was identified near the tegument cluster (*Figure 1C*). The top markers were Smp_096390, a 16-kDa calcium-binding protein (CaBP) and *p25 alpha* (Smp_097490, a marker for ciliated neurons in adult worms, *Wendt et al., 2020*; *Figure 3A*). In situ hybridisation revealed CaBP and *p25 alpha* expression in the ciliary plates; epithelial cells with thousands of motile cilia used for swimming (*Figure 3G* and *Figure 3—figure supplement 2*). Both genes were also expressed in six previously undescribed cells that sent a projection out to the exterior (between the first and second tiers of ciliary plates as seen with phalloidin staining, *Figure 3G*) and whose nuclei sat below the body wall musculature (*Figure 3—videos 2; 3*; *Figure 3—figure supplement 2F*). Given the association of *p25 alpha* with ciliated neurons in adults (*Wendt et al., 2020*), expression in the submuscular cells suggests they project cilia into the external

environment and may be sensory cells. The expression of CaBP and *p25* in both the ciliary plates and submuscular cells suggests that these anatomical structures are functionally linked. GO terms for this cluster were related to 'mitochondrial transmembrane transport' and 'microtubule-based process' (*Supplementary file 1g*), and STRING analysis showed two interaction networks: one associated with 'carbon metabolism and mitochondria envelope', the second associated with 'cytoplasmic dynein complex' and 'dynein light chain superfamily' (*Figure 3—figure supplement 4*). Further plate-based scRNA-seq of 21 individual ciliary plates confirmed the high expression of CaBP (Smp_096390) and *p25 alpha* (Smp_097490), with more than twofold higher expression of CaBP than any other gene by median TPM (*Supplementary file 1l*). Although the ciliary plates expressed *p25* (a ciliated neuron marker in the adult, *Wendt et al., 2020*), they did not express many of the typical neural markers, for example *complexin* and *7B2* (*Figure 3—figure supplement 6*).

## Tegumental cells expressed *MEG6* and are located in the posterior two-thirds of the larva

We identified a cluster of 366 cells as tegumental based on the expression of markers for tegument (skin) in the adult worm (*Wendt et al., 2020*), for example genes encoding *dynein light chain* (Smp_302860, Smp_200180), a DUF3421 domain-containing protein (Smp_331910), and micro exon gene *MEG6* (Smp_163710). Two other dynein light chain genes were top markers (Smp_201060, Smp_312630), and both a tetraspanin (Smp_303290) and *Gelsolin* (Smp_008660) were specific to this cluster. *Meg6* (Smp_163710) was expressed in 66% of the cells in this cluster; ISH showed 46 *Meg6*[+] cells, and the nuclei were in the posterior two-thirds of the larva. The nuclei sat below the BWM, and cytoplasmic protrusions reached between muscle filaments and formed the epidermal ridges between the ciliary plates (*Figure 4C, D*). These cells correspond to the epidermal ridge cells described by *Pan, 1980*. A second tegument marker, a dynein light chain (Smp_200180), showed overlapping expression with *Meg6* (*Figure 4—video 4*). Enriched GO terms for tegument genes included 'microtubule-based process' and 'protein folding' (*Supplementary file 1g*). STRING analysis revealed a network of 25 genes associated with protein localisation to the ER, folding, export and post-translational protein targeting to the membrane (*Figure 4—figure supplement 1*).

## Spatial localisation of top marker transcripts identifies a simple protonephridial system

There were 536 cells in the protonephridial cluster, expressing genes common to excretory systems in other animals (*Gąsiorowski et al., 2021*), for example, the structural protein *nephrin* (Smp_130070) and the transcription factor *odd-skipped-related* (Smp_152210). Eighty percent of the cells expressed the protonephridia marker Smp_335600 (ShKT-domain-containing protein) (*Nanes Sarfati et al., 2021*; *Figure 4A*). Smp_335600 was expressed around six nuclei in a bilaterally symmetrical pattern (*Figure 4Ei*; *Figure 4—video 1*). On each side, Smp_335600 transcripts were distributed along the s-shaped path of an excretory tubule, passing its nucleus, and connecting the anterior and posterior flame cells (*Figure 4Ei*). Within the flame cells, expression was detected around the nuclei rather than the barrel (*Figure 4Eii*). The four ciliated flame cells and two excretory ducts comprise the excretory system of the miracidium. Accordingly, STRING (*Figure 4—figure supplement 1*) and GO analyses (*Supplementary file 1g*) of this cluster confirmed enrichment for terms including 'cilium organisation', 'axoneme assembly', 'cytoskeleton organisation', and 'microtubule-based process'.

## Two distinct parenchyma clusters predicted in the miracidium

We identified two clusters of cells (1093 cells from Parenchyma 1 and 605 cells from Parenchyma 2) that were transcriptionally similar to the parenchymal cells found in later developmental stages; for example, they express hypothetical protein (Smp_063330), *cathepsin B-like cysteine proteinase* (Smp_141610), and *serpin* (Smp_090080) (*Diaz Soria et al., 2020*; *Wendt et al., 2020*; *Diaz Soria et al., 2024*). All top marker genes for Parenchyma 1 were also expressed in Parenchyma 2 but at lower levels, whereas the DUF2955 domain-containing protein (Smp_191970) was uniquely expressed in Parenchyma 2 (*Figure 4A*). A gene encoding a Zinc transporter Slc39a7 (Smp_318890) was a top marker for both clusters, with expression confirmed in situ within seven to nine cells; two of which were located anteriorly on either side of the brain and five to seven located posterior to the brain (*Figure 4B, F*). These parenchyma cells have long cytoplasmic protrusions that reach between all the

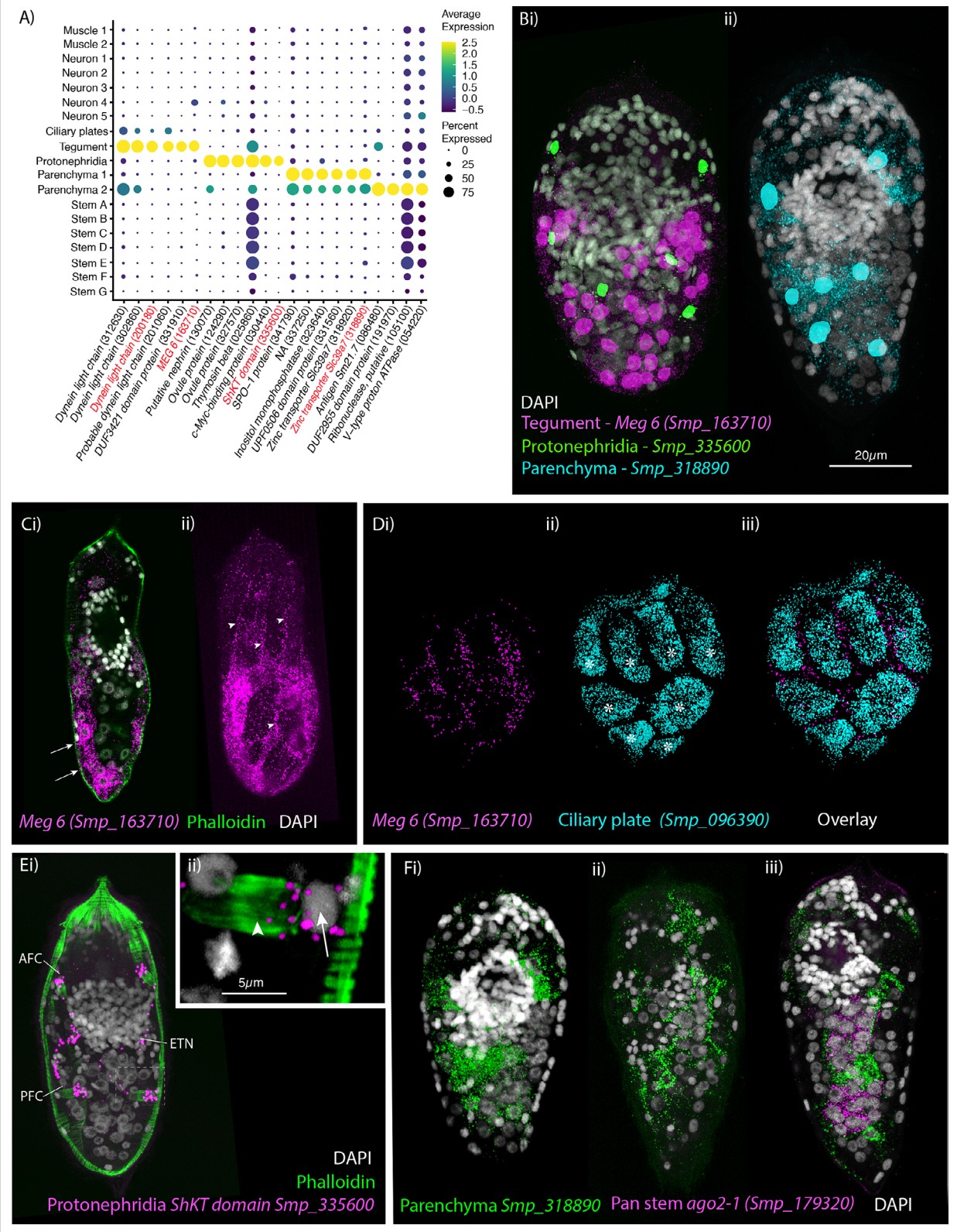

**Figure 4.** Identification of the tegument, protonephridia, parenchymal cells in the miracidium. (**A**) Expression profiles of cell marker genes specific or enriched in these cell-type clusters. Genes validated by in situ hybridisation are in red. Gene identifiers shown in parenthesis but with 'Smp_' prefix removed for brevity. Gene expression has been log-normalised and scaled using Seurat(v. 4.3.0). (**B**) Segmentation of nuclei of the cells expressing the marker genes for: (**i**) protonephridia and tegument (multiplexed) show that six cells express Smp_335600 (*ShKT-domain protein*, a marker for

*Figure 4 continued on next page*

*Figure 4 continued*

protonephridia), 46 cells express *Meg-6* (a marker for tegument) (segmentation of 1 larva), and (**ii**) seven cells that express an uncharacterised gene, Smp_318890 (a marker for parenchymal cells)(segmentation of 1 larva). (**C, D**) The tegument marker *Meg-6* shows expression around nuclei in the posterior two-thirds of the larvae. The nuclei are below the body wall muscle, and cytoplasmic protrusions reach between muscle filaments (arrows) and form the epidermal ridges (arrowheads) between the ciliary plates (which are visible in **Dii, iii** expressing the ciliary plate marker Smp_096390) (asterisks). 100% of individuals examined, $n > 30$. (**E**) The protonephridial marker Smp_335600 shows 'S'-shaped expression with transcripts extending from the nucleus of the anterior flame cell (AFC) along the excretory tubule and its nucleus (ETN) to the posterior flame cell (PFC), (**Eii**) it is expressed around the nuclei (arrow) rather than the barrels (arrowhead) of the flame cells. 100% of individuals examined, $n > 30$. (**Fi**) The pan-parenchymal marker, Smp_318890, was expressed in two anterior cells (one on either side of the brain) and five to seven cells posterior to the brain. (**Fii**) These cells have long cytoplasmic protrusions that reach between all the other cells, (**Fiii**) including the *ago2-1*+ stem cells. 100% of individuals examined, $n > 30$. Scale shown in B also applies to C, D, Ei, and F.

The online version of this article includes the following video and figure supplement(s) for figure 4:

**Figure supplement 1.** STRING analysis of tegument and protonephridia clusters.

**Figure supplement 2.** STRING analysis of parenchyma clusters.

**Figure supplement 3.** Gene ontology (GO) analysis of marker genes identified in the miracidia cell clusters.

**Figure 4—video 1.** Maximum intensity projection from confocal z-stack of wholemount in situ hybridisation (WISH) of protonephridia marker *ShKT-domain-containing protein* (Smp_335600) (magenta), counterstained with DAPI (white).

https://elifesciences.org/articles/95628/figures#fig4video1

**Figure 4—video 2.** Confocal z-stack of wholemount in situ hybridisation (WISH) of markers for parenchyma 1 and 2 (Smp_318890) (green) and stem cells *ago2-1* (Smp_179320) (magenta), counterstained with DAPI (white) shows the long cytoplasmic protrusions of the parenchymal cells reaching between the stem and other cells.

https://elifesciences.org/articles/95628/figures#fig4video2

**Figure 4—video 3.** Maximum intensity projection from confocal z-stack of wholemount in situ hybridisation (WISH) of marker for parenchyma 1 and 2 (Smp_318890) (green) counterstained with DAPI (white) shows the distribution of the parenchymal cells and their reach in the intercellular space of the larva.

https://elifesciences.org/articles/95628/figures#fig4video3

**Figure 4—video 4.** Confocal z-stack of wholemount in situ hybridisation (WISH) of markers for tegument *Meg6* (Smp_163710) (magenta) and *dynein light chain* (Smp_200180) (cyan), counterstained with phalloidin (green) and DAPI (white) shows the nuclei sit below the body wall muscle, and cytoplasmic protrusions reach between muscle filaments and form the epidermal ridges between the ciliary plates.

https://elifesciences.org/articles/95628/figures#fig4video4

other cells, filling the intercellular space (*Figure 4Fii, iii*; *Figure 4—videos 2; 3*). *Pan, 1980* called these the interstitial cells and, observing abundant glycogen particles and lipid droplets, proposed they function as sources of energy for the non-feeding larva and as nurse cells for the stem cells during the first few days post-infection of the snail. We performed ISH for two Parenchyma 2 markers, Smp_191970 and Smp_320330, but the first showed no detectable signal, and the second had broad expression across all cells with no specific overlap with the pan-parenchymal marker Smp_318890. Therefore, by ISH, we could only identify parenchymal cells in general and could not identify the two clusters in situ. STRING analysis for the Parenchyma 1 showed enrichment in networks for 'carbo-hydrate metabolic process' and 'hydrolase', and for Parenchyma 2: 'phagosome', 'mTOR signalling pathway', 'mitochondrial inner membrane', and 'proton-transporting two sector atpase complex' (*Figure 4—figure supplement 2*). Analysis of GO annotations was able to distinguish between the two parenchyma subpopulations, with Parenchyma 1 genes enriched for 'extracellular region' and 'endopeptidase inhibitor activity', and Parenchyma 2 genes enriched for processes that are 'dynein complex' and 'microtubule-based process' (*Supplementary file 1g*; *Figure 4—figure supplement 3*).

## Two major stem cell populations in the miracidium that clustered by sex

We identified stem cells based on the expression of known stem cell markers for the mother sporo-cyst stage, for example *ago2-1* (Smp_179320), *nanos-2* (Smp_051920), *fgfrA* (Smp_175590), *fgfrB* (Smp_157300), *klf* (Smp_172480), and *hesl* (Smp_024860) (*Wang et al., 2013*; *Wang et al., 2018*; *Figure 5A*). The mother sporocyst has been reported to have three stem cell populations: Kappa (*klf*+, *nanos-2*+), Delta (*fgfrA*+, *fgfrB*+, and *nanos-2*+), and Phi (*fgfrA*+, *fgfrB*+, and *hesl*+) based on scRNA-seq of 35 stem cells (*Wang et al., 2018*). *Wang et al., 2018* suggest that the Kappa population serve as 'embryonic' stem cells that give rise to both the Delta and Phi populations; Delta cells generate somatic tissues and Phi produce temporary larval tissues, including tegument. In the miracidium, we

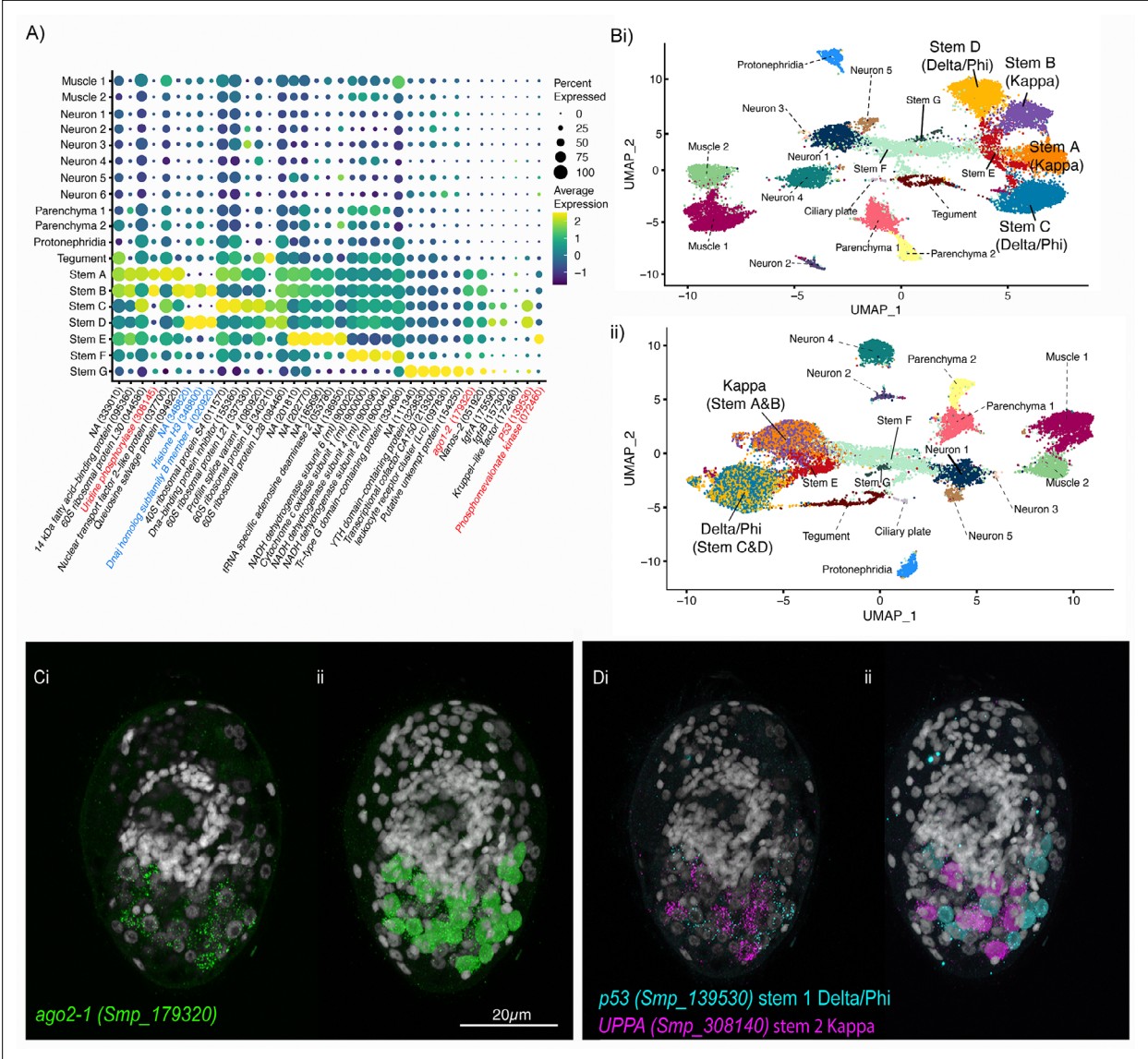

**Figure 5.** Two defined populations of stem cells cluster by sex. (**A**) Expression profiles of cell marker genes that are specific or enriched in the stem cell clusters. Genes specific to the W (female-specific) sex chromosome are highlighted in blue. Genes validated by ISH are marked in red. Gene identifiers shown in parenthesis but with 'Smp_' prefix removed for brevity. Gene expression has been log-normalised and scaled using Seurat(v. 4.3.0). (**Bi**) UMAP including all genes shows that there are two Delta/Phi and two Kappa stem clusters, (**ii**) UMAP showing that removal of all genes specific to the W and Z sex chromosomes results in one Delta/Phi and one Kappa cluster, indicating that the stem cells are transcriptionally different in male and female miracidia due to the expression of sex-linked genes. (**C, D**) Multiplexed ISH showing three stem cell markers simultaneously in the same individual: *ago2-1* (Smp_179320) (pan-stem), *p53* (Smp_139530) (Delta/Phi), and *Uridine phosphorylase A* (*UPPA*, Smp_308140) (Kappa). (**Ci**) *Ago2-1* expression reveals stem cells lateral and posterior to the brain (Optical section). 100% of individuals examined, *n* > 30. (**ii**) MIP with segmentation of the 23 *ago2-1*+ cells. (**Di**) *P53* and *UPPA* expression shows the two stem cell populations intermingled in 100% of individuals examined, *n* > 30. (**ii**) MIP and segmentation reveal there are more *p53*+ cells than *UPPA*+ cells (in this larva,15 are *p53*+ and 9 are *UPPA*+). Scale shown in C also applies to D.

The online version of this article includes the following video and figure supplement(s) for figure 5:

**Figure supplement 1.** Expression of known stem cell markers in miracidia cells.

**Figure supplement 2.** The effect of sex-linked genes on cell clustering.

**Figure supplement 3.** Identification of new clusters after removing Z-specific region (ZSR) and W-specific region (WSR) genes.

**Figure supplement 4.** Classification of cells based on percentage of reads mapping to W-specific region indicates incomplete dosage compensation of Z-specific region genes.

**Figure supplement 5.** 5-ethynyl-2'-deoxyuridine (EdU) labelling shows that there are no dividing cells during the free-swimming miracidia stage.

*Figure 5 continued on next page*

*Figure 5 continued*

**Figure supplement 6.** STRING analysis of the top 100 marker genes for the two stem cell populations in the miracidium (combining the female and male cells).

**Figure supplement 7.** Differential gene expression between the two stem cell populations, Delta/Phi and Kappa.

**Figure 5—video 1.** Confocal z-stack of wholemount in situ hybridisation (WISH) of markers for stem clusters Delta/Phi *p53* (Smp_139530) (cyan), Kappa *UPPA* (Smp_308145) (green), and stem E *Phosphomevalonate kinase* (Smp_072460) (magenta), counterstained with DAPI (white), shows there are no cells that are *pmk*-positive but negative for *UPPA* and *p53*.

https://elifesciences.org/articles/95628/figures#fig5video1

**Figure 5—video 2.** Confocal z-stack of wholemount in situ hybridisation (WISH) of markers for stem clusters Delta/Phi *p53* (Smp_139530) (cyan), Kappa *UPPA* (Smp_308145) (magenta), and the pan-stem marker *ago2-1* (Smp_179320) (green), counterstained with DAPI (white), shows there are no cells that are *ago2-1* positive but negative for *UPPA* and *p53* that could be Stem F and G cells.

https://elifesciences.org/articles/95628/figures#fig5video2

identified seven stem cell clusters by expression of *ago2-1* (10,686 cells): Stems A and B are *klf*$^+$ and *nanos-2*$^+$ (i.e. Kappa-like); Stems C and D are *nanos-2*$^+$, *fgfrA*$^+$, *fgfrB*$^+$, and *hesl*$^+$ (i.e. they resembled both Delta and Phi) (**Figure 5A**, **Figure 5—figure supplement 1**); as well as Stems E, F, and G (**Figure 5B**).

Working with the latest version of *S. mansoni* genome (v10), where the sex chromosomes Z and W have been resolved (**Buddenborg et al., 2021**), we were able to determine that the Delta/Phi- and Kappa-like populations are each composed of two well-separated clusters that can be distinguished based on the expression of sex-linked genes (**Figure 5A**; **Figure 5—figure supplement 2**). *S. mansoni* has seven pairs of autosomes and one pair of sex chromosomes; males are homogametic (ZZ) and females heterogametic (ZW) (**Buddenborg et al., 2021**). The sex chromosomes are composed of sex-specific regions that are unique to each chromosome flanked by pseudoautosomal regions that are common to both Z and W. To unveil the genes contributing to the sex-driven clustering we removed in turn different sex chromosome regions from the analysis. When all Z-specific region (ZSR) and W-specific region (WSR) genes were removed from the dataset and the cells reclustered, the two Delta/Phi-like and two Kappa-like clusters each collapsed into single stem cell clusters, giving five stem cell clusters in total (i.e. Delta/Phi, Kappa, E, F, and G; **Figure 5B**, **Figure 5—figure supplement 3**). Removing from the dataset genes located solely on the WSR ($n \approx 35$), the ZSR ($n \approx 900$), or just the ~35 pairs of genes with homologous copies on the WSR and ZSR did not impact the overall UMAP topology, that is in all three scenarios the stem cell clusters remained separated by sex (**Figure 5—figure supplement 2**). ZSR or WSR expression is therefore sufficient on their own to split each of these stem clusters in two (**Figure 5—figure supplement 2**).

The unexpected subclustering of Delta/Phi and Kappa stem cells by sex could be either due to the contribution to each cell's transcriptome by WSR genes (in females) that have diverged from their ZSR homologues, or by differences in expression of ZSR genes between male or female cells (see GO analysis of ZSR and WSR genes, **Supplementary file 1m**). Given the small number of WSR genes, differences in ZSR genes – caused by a difference in copy-number – seems likely to play a larger part. In many animals, dosage compensation equalises gene expression across the sex chromosomes that are unevenly distributed between the sexes. To investigate whether there is an incomplete or a lack of dosage compensation in the ZSR genes in the *S. mansoni* miracidium, we classified all cells based on the expression of WSR genes (i.e. only found in female individuals), so each cell was designated as W$^+$ (likely female) or W$^-$ (likely male) (**Figure 5—figure supplement 2A, B**, **Figure 5—figure supplement 4A, B**). The difference in ZSR and WSR gene expression was most pronounced in the Delta/Phi- and Kappa-like cells, shown by the separation of W$^+$ and W$^-$ cells (**Figure 5—figure supplement 4B**). However, this analysis revealed other tissue-specific sex differences in ZSR gene expression for example in the protonephridia, which was evident through segregation of W$^+$ and W$^-$ cells within that cluster. We found that W$^+$ and W$^-$ cells were well mixed in the majority of the remaining clusters (**Figure 5—figure supplement 4B**). Male cells have two copies of the ZSR (ZZ) and produce higher coverage across genes in this region compared to female cells, which have one ZSR and one WSR (ZW). These findings suggest that sex-linked gene dosage is not fully compensated in the miracidium, and there may be tissue-specific variation.

To identify the stem cells in situ, we investigated the expression of the pan-stem marker *ago2-1*. *Ago-1*$^+$ cells were distributed lateral and posterior to the brain (**Figure 5Ci**) and there was variation

in the number between larvae, ranging from 20 to 29 (segmentation of *Ago2-1*[+] nuclei in 5 larvae, median = 23, mean = 23.4) (i.e. ~7% of cells that make up the larva are stem cells). Our count is slightly higher than previous estimates of 10–20 stem/germinal cells (*Pan, 1980*; *Wang et al., 2013*). Cells in the Delta/Phi-like cluster expressed a canonical *p53 tumour suppressor family gene* (*p53-1*, Smp_139530) as a top marker; in situ expression showed 12–14 *p53-1*[+] cells (segmentation of three larvae) (*Figure 5D*). Cells in the Kappa-like cluster expressed *Uridine phosphorylase A* (*UPPA*, Smp_308145) as a top marker; in situ expression showed eight to nine *UPPA*[+] cells, and they were located amongst the *p53-1*[+] cells (segmentation of three larvae) (*Figure 5D*). We were not able to detect cells from the Stem E, F, and G clusters in the miracidium using ISH as there were no predicted marker genes that were unique to these clusters. Although *Phosphomevalonate kinase* (*pmk*) (Smp_072460) was a predicted marker for the Stem E cluster (*Figure 5A*), when multiplexed as *p53/UPPA/pmk*, no *pmk*[+]-only cells could be found (*Figure 5—video 1*). Likewise, multiplexing *ago2−1/p53/UPPA* revealed no *ago2-1*[+]-only cells (*Figure 5—video 2*).

Overall, we could spatially validate two stem cell populations within the miracidia: Delta/Phi and Kappa. Cells in Stem E, F, and G in silico clusters might be stressed/damaged/dying cells or cells in transcriptionally transitional states (e.g. some *complexin* expression was also found in a subpopulation of Stem F).

A lack of EdU incorporation into miracidia cells (*Figure 5—figure supplement 5*) indicated that DNA replication does not take place in the stem or somatic cells during this free-swimming stage. As the first stem cells start proliferating in the mother sporocyst ~20 hr post-infection (*Wang et al., 2013*), this suggests that both stem cell populations pause cell division for at least 26 hr.

Many top markers for the Delta/Phi and Kappa stem clusters were ribosomal proteins (*Figure 5A*). STRING analysis showed that most of the top 100 markers of each cluster both formed large single networks enriched for genes involved in translation, ribosome biogenesis, and gene expression, and this was particularly striking in the Delta/Phi cluster (*Figure 5—figure supplement 6*). Amongst the top genes differentially expressed between Kappa and Delta/Phi (*Figure 5—figure supplement 7A*; *Supplementary file 1n*) were six transcription factors upregulated in Delta/Phi: an orthologue of *lin-39* (Smp_247590), *tiptop* (Smp_180690), *hes-related* (Smp_024860), *p53-1* (Smp_139530), *Zfp-1* (Smp_145470), and a putative *homothorax homeobox protein/six-3–1* (Smp_147790). The simultaneous co-expression of multiple transcription factors indicates a multipotent progenitor stage (*Kumar et al., 2017*). Enriched in, or specific to, Kappa cells were genes involved in glycolysis and lipid metabolism, that is malate dehydrogenase (Smp_129610), 14 kDa fatty acid-binding protein (Smp_095360) and a very long-chain fatty acid elongase (Smp_051810). GO analysis of differentially expressed genes (*Supplementary file 1o*) revealed that the most significantly upregulated genes in Delta/Phi were related to the structural component of the ribosome and translation; these are the first markers of a pre-activation stage in stem cells (*Shin et al., 2015*). Whereas GO analysis showed that upregulated genes in Kappa are involved in transcription and metabolism (*Figure 5—figure supplement 7B*), which indicated quiescent rather than primed and activated stem cells (*Llorens-Bobadilla et al., 2015*).

## One miracidia stem population contributes to somatic tissues and the other to the germline

The Delta/Phi stem population simultaneously co-expressed three transcription factors that are known to be specific to the flatworm tegument/epidermal lineage, that is *p53-1*, *zfp-1*, and *six3-1* (*Cheng et al., 2018*; *Collins et al., 2016*; *Wendt et al., 2018*). This suggests that the Delta/Phi stem population may have been the origin of the miracidia tegument cells and may be the source of new tegument cells in the mother sporocyst stage. To explore the fates of the two miracidia stem populations, we investigated the dynamics of their gene expression in stem and tegument cells across the transition from miracidia to sporocyst. To do this, we combined single-cell data from these developmental stages and cell types (this study and *Diaz Soria et al., 2024*) and used RNA velocity analysis (*La Manno et al., 2018*; *Bergen et al., 2020*) to estimate relative induction/repression and timing of gene expression in this dataset.

A UMAP plot of the combined clustering shows the gene expression dynamics of the miracidia and sporocyst tegument and stem cells (*Figure 6*) (batch-corrected samples showed similar results, *Figure 6—figure supplement 1*). The velocity field and latent time representations indicate the

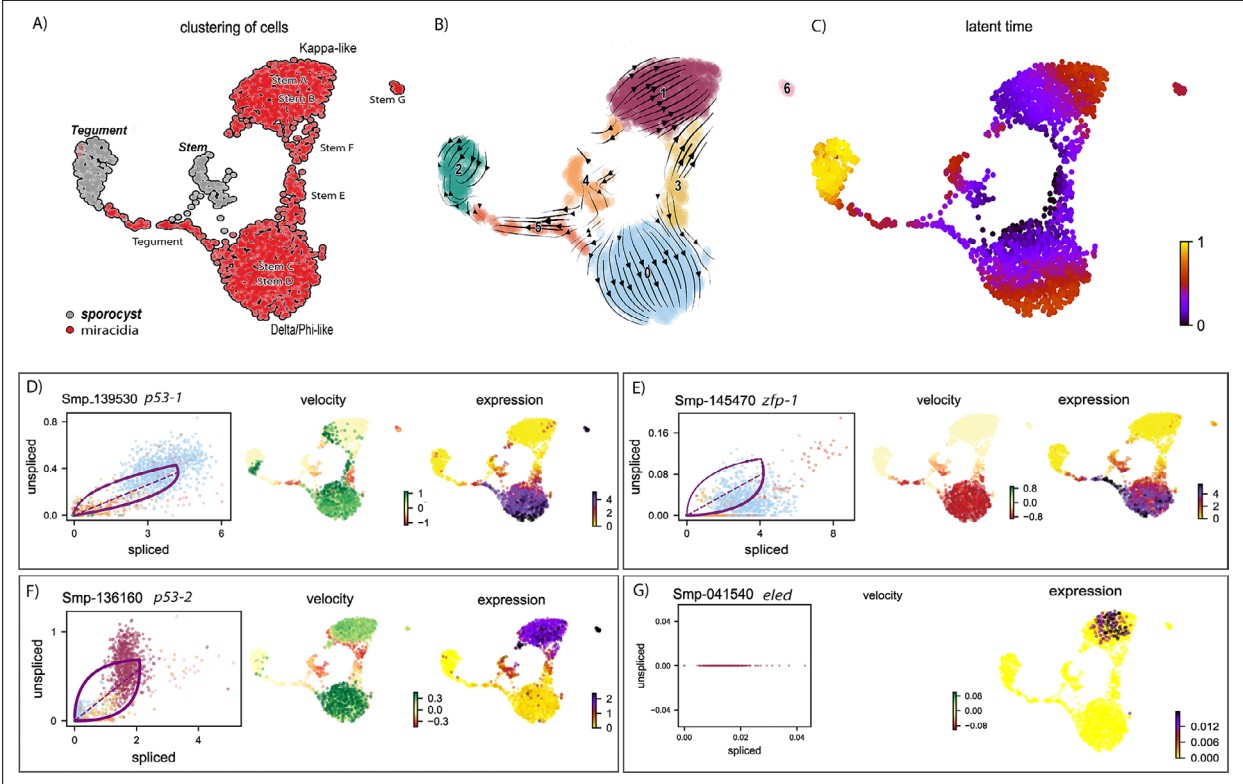

**Figure 6.** RNA velocity analysis of stem and tegument cells from the miracidium and sporocyst shows lineage-specific gene dynamics. (**A**) UMAP shows the life cycle stage origin of cells and cell cluster identity from Seurat analysis. (**B**) RNA velocity analysis flow field shows the generalised direction of RNA velocity. (**C**) Latent time analysis shows an estimated temporal relationship between cells. (**B** and **C**) are based on the expression of both spliced and unspliced transcripts, and their expression dynamics across cells and genes. The phase plot, velocity, and expression were calculated for (**D**) *p53-1*, (**E**) *Zfp-1*, (**F**) *p-53-2*, and (**G**) *eled* (no batch correction). Gene expression has been normalised using counts per million (CPM Expanded and re-phrased) and log-transformed using scvelo(v. 0.2.4). For all of (**D–G**), the phase plot (left plot) shows the proportion of spliced and unspliced transcripts in each cell, where each point is a cell and is coloured by the clusters in (**B**). The purple almond-shape overlaid represents the processes of transcription, splicing, and degradation, where this can be modelled. The dashed line shows the estimated steady state where RNA transcription is constant. The middle panel shows the RNA velocity, which for each gene is based on how the observations deviated from the estimated steady state towards induction or repression. The right panel shows gene expression. *p53-1* and *zfp-1* are predominantly expressed in the Delta/Phi-like miracidia stem cells and some miracidia tegument, and velocity indicates active expression of *p53-1* but downregulation of *zfp-1* in the stem cells. *p53-2* is most highly expressed in the Kappa-like miracidia stem cells, and velocity indicates this gene is being actively transcribed. *eled* expression is very low and only spliced transcripts have been detected, but is generally restricted to the Kappa-like miracidia stem cells.

The online version of this article includes the following figure supplement(s) for figure 6:

**Figure supplement 1.** RNA velocity analysis of stem and tegument cells from the miracidium and sporocyst (with batch correction) shows lineage-specific gene dynamics.

**Figure supplement 2.** RNA velocity analysis of stem and tegument cells from the miracidium and sporocyst (without batch correction).

**Figure supplement 3.** RNA velocity analysis of stem and tegument cells from the miracidium and sporocyst (without batch correction).

**Figure supplement 4.** RNA velocity analysis of all tissue types from the miracidium and sporocyst shows lineage-specific gene dynamics.

**Figure supplement 5.** RNA velocity analysis of all tissue types from the miracidium and sporocyst shows the dynamics of described germline-related genes.

**Figure supplement 6.** RNAvelocity analysis of all tissue types from the miracidium and sporocyst shows lineage-specific gene dynamics.

progression of some miracidia Delta/Phi stem cells towards the miracidia tegument cells and a high speed of differentiation was inferred from the miracidia tegument cells to the sporocyst tegument (*Figure 6B, C*). This suggests that the miracidia tegument cluster may contain cells that range from post-mitotic stem cell progeny to tegument progenitors to terminally differentiated tegumental cells.

Investigations into tegument development in adult schistosomes has revealed key regulators and genes associated with different development states (*Collins et al., 2016*; *Wendt et al., 2018*; *Wendt et al., 2022*). We examined the dynamics of these genes in the combined miracidia/mother sporocyst

dataset using RNA velocity and found that the velocities of these genes show different temporal dynamics. *P53-1* (Smp_139530) was induced in the Delta/Phi stem cells but repressed in some miracidia tegument cells and expression decreased in a continuum across the miracidia tegument cells (*Figure 6D*). The flatworm-specific transcription factor *Zfp-1* (Smp_145470) showed a similar expression pattern to *p53-1*, except that the predominance of spliced copies (*Figure 6E*) suggested that transcription happened earlier, during embryogenesis. A second zinc finger protein, *Zfp-1-1* (Smp_049580) was expressed in tegumental cells more than stem cells, which suggested a role in stem cell progeny and tegument progenitors (*Figure 6—figure supplement 2*). In adults, tegument cells are renewed continuously by a population of *tsp2+* (Smp_335630) progenitor cells that fuse with the tegument (*Wendt et al., 2018*). In the miracidia, *tsp2* was expressed in the Delta/Phi stem cells as well as tegument cells (*Figure 6—figure supplement 2*), suggesting earlier expression than in the adult, or that Delta/Phi are progenitor cells. Several markers of mature tegument cells in the adult were expressed in the miracidia and sporocyst tegument cells but not in the stem cells, for example *annexin* (Smp_077720), *calpain* (Smp_241190), and *endophilin B* (Smp_003230) (*Figure 6—figure supplement 2*). *Annexin* is specific to the tegument cells, and the phase plot showed the accumulation of mature transcripts in the sporocyst tegument, while there was evidence of recent transcription in the miracidia tegument (*Figure 6—figure supplement 2*). A putative orthologue of a planaria transcription factor *six-3-1* (Smp_147790), a marker for early epidermal progenitors (*Cheng et al., 2018*) was expressed in the Delta/Phi stem cells and miracidia tegument (*Figure 6—figure supplement 2*). All of these genes showed little expression in the Kappa stem population (*Figure 6—figure supplement 2*). The expression of these known tegument/epidermal development and lineage-specific genes in the Delta/Phi stem cluster, and the indication that *zfp-1* transcription occurred pre-hatching, suggested that this stem population was the origin of the miracidia tegument cells and is the source of the tegumental lineage. This, together with the observation that more than 80% of protein-coding genes in the genome are detected in these miracidia data, suggests that similar molecular programmes for the development of tissues in the miracidia and mother sporocyst may be redeployed multiple times across the life cycle.

As well as allowing us to interrogate the dynamics of known marker genes (*Figure 6—figure supplement 2*), the RNA velocity analysis also revealed genes previously unknown to have dynamic behaviours across these stages: these included genes upregulated from miracidia tegument to sporocyst tegument, for example *epidermal growth factor receptor* (Smp_035260), genes downregulated from miracidia tegument to sporocyst tegument, for example *galectin domain* (Smp_140590), and dynamic behaviour across stem cells and tegument, for example *unc44/ankyrin 2,3* (Smp_145700) (*Figure 6—figure supplement 2*, *Supplementary file 1p*).

In the Kappa stem population, there was transcription and expression of genes detected in germline stem cells and germ cells in the intra-mammalian stages of the life cycle (*Wang et al., 2018*; *Li et al., 2021*). A second P53 homologue, *p53-2* (Smp_136160), was highly expressed and actively transcribed in Kappa cells (*Figure 6F*). In adults, this gene is enriched in reproductive organs and has a genotoxic stress response that is thought to defend the integrity of the genome by eliminating germ cells that have acquired mutations (*Wendt et al., 2022*). Also expressed solely in Kappa cells (though some at low levels) are the germline-specific regulatory genes *eled* (Smp_041540) (*Figure 6G*), *onecut1* (Smp_196950), *irx* (Smp_063520), *akkn* (Smp_131630), and *boule* (Smp_144860) (expressed in the female Kappa cells and Stem E, *Figure 6—figure supplement 5*; *Wang et al., 2018*; *Li et al., 2021*), as well as *klf* (Smp_172480) (*Figure 6—figure supplement 3*), a homologue of KLF4, which is an important transcriptional regulator governing pluripotency (*Takahashi et al., 2007*).

Together, these data suggest that the Delta/Phi population is the origin of the tegumental lineage and likely contributes to extra-embryonic somatic tissues during intra-molluscan development, while the Kappa population likely contains pluripotent cells that will contribute to the development of cercariae and the germline in intra-mammalian stages (*Figure 7*).

## Discussion

Schistosomes are complex metazoan parasites, with adult worms of *S. mansoni* characterised into 68 transcriptionally distinct cell clusters (*Wendt et al., 2020*). Understanding the genes involved in the development and maintenance of these cell types is a promising approach for identifying novel therapeutic targets (*Wendt et al., 2020*). Here we have shown that the miracidium larva is composed

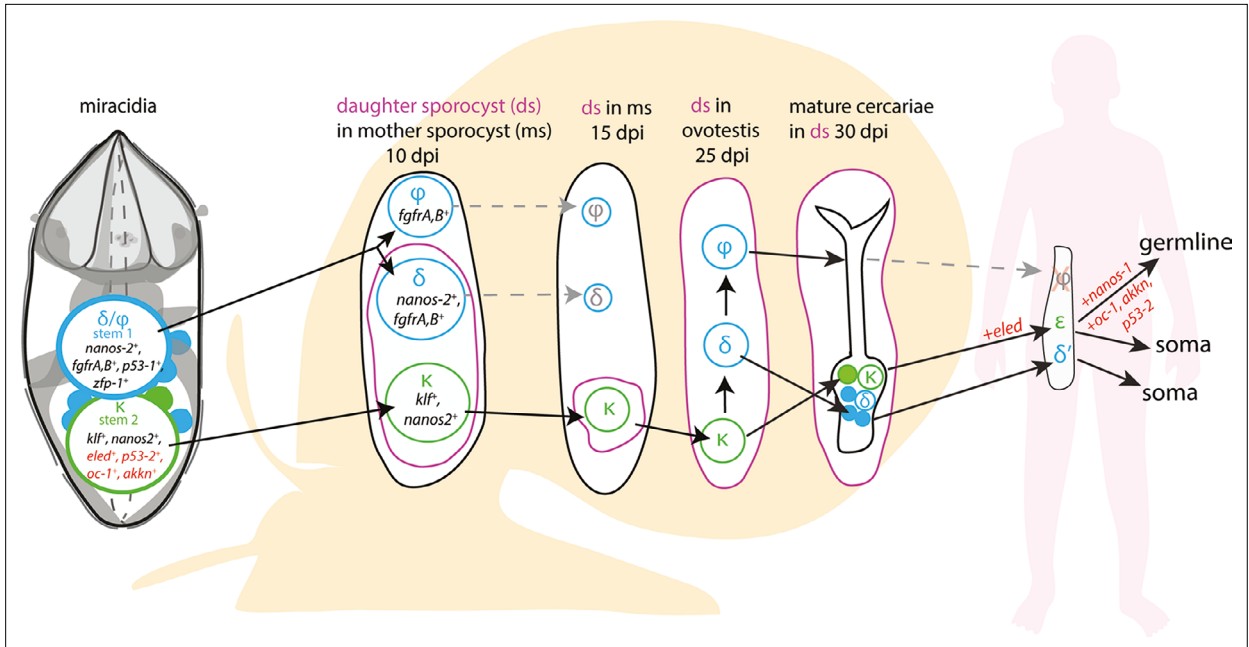

**Figure 7.** A model for the fate of the two stem populations in the miracidium. Adding single-cell data for miracidia stem cells (this study) to existing stem cell scenarios on *Schistosoma mansoni* development (*Wang et al., 2018*; *Li et al., 2021*) show the continuum of the Kappa (κ) population from the miracidium, through the intra-molluscan stages to the juvenile and adult stages inside the mammalian host. *Wang et al., 2018* proposed that the κ cells give rise to epsilon (ε), *eled⁺*, cells in the juvenile primordial testes, ovaries, and vitellaria (germline), as well as in a gradient increasing towards the posterior growth zone (soma). They suggested that germ cells may be derived from ε-cells early in juvenile development, and *eled* is the earliest germline marker yet identified in schistosomes. This led to the idea that *S. mansoni* does not specify its germline until the juvenile stage (*Wang et al., 2018*) and a germline-specific regulatory programme (including *eled, oc-1, akkn, nanos-1, boule*) was identified in intra-mammalian stages (*Wang et al., 2018*; *Li et al., 2021*). We show expression of these genes in κ stem cells in the miracidium. This suggests that after ~6 days of embryogenesis, at hatching of the miracidium, the cells that may contribute to the germline might already be segregated into the κ population and the molecular regulatory programme that differentiates somatic (delta/phi, δ/φ) and germ cell (κ) lineages is present. Furthermore, as *p53-2* plays a genotoxic stress response role in adult reproductive cells (*Wendt et al., 2022*), its expression in κ cells in the miracidia is another line of evidence that indicates that this population may contain the pluripotent stem cells that likely give rise to the germline.

of ~365 cells. Compared to later life cycle stages, where the cercariae is made up of ~1600 cells, the schistosomulum ~900 (*Diaz Soria et al., 2020*) and the adults tens of thousands of cells (unpublished data), the miracidium is a simple stage of the life cycle. We identified and spatially resolved 19 transcriptionally distinct cell types, revealing the relative contribution of each tissue type to the larva (*Figure 8*). The miracidium, therefore, with its diversity of differentiated cells present in relatively low numbers, makes a simple and tractable system with which to gain a fundamental understanding of the biology and spatial architecture of schistosome cells, and their transcriptomes.

Nineteen cell types may be a conservative characterisation, but at this resolution most cell types formed clearly defined clusters. There was sufficient sensitivity to detect cell types with as few as six cells per larva, for example, Neurons 2 and 3, and the protonephridial system. Some rarer cell types are embedded in existing clusters, particularly in Neuron 1, and a complete neural cell atlas and a study of the putative photoreceptor cells is in progress (Rawlinson et al., in prep). Some cell types were preferentially recovered, for example the stem cells; ISH revealed that they comprise only 7% of the total cell number in the larva, yet they accounted for ~52% of the scRNA-seq data (*Supplementary file 1q*). Preferential recovery may be due to differences in the robustness of different cell types to dissociation, cell sorting, and/or the microfluidics of the 10X GEM capture process. The ciliary plates, for instance, were only found in sufficient numbers to be defined as a cluster in unsorted samples. Therefore, there are clear benefits to including unsorted cell samples to discover the diversity of cell types in a tissue, organ, or organism, especially when cell types may be fragile, sticky, or unusually shaped. Even with the inclusion of unsorted cells, some known cell types were not captured, that is the penetration glands. With three gland cells per larva and a theoretical 56× coverage of each cell, we would have expected ~168 gland cells in the dataset. These cells were easily identified

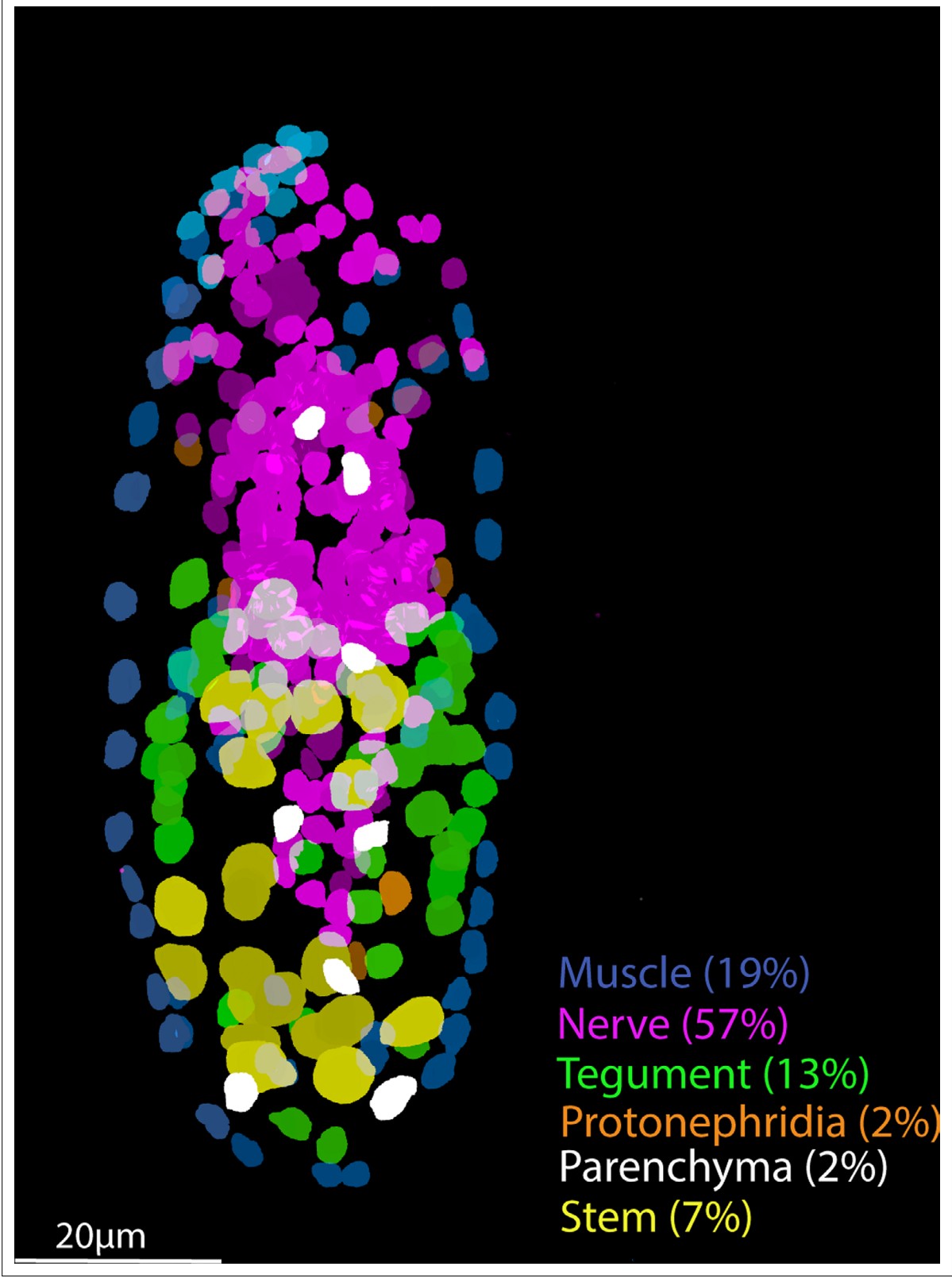

**Figure 8.** Tissue-level segmentation of a miracidium reveals the contribution of each tissue to this simple larva. In situ hybridisation using HCR (hybridisation chain reaction) enabled multiple tissue-level marker genes to be visualised simultaneously in the same larva. The nuclei of the cells of each tissue type were manually segmented using TrakEM2 in ImageJ (*Cardona et al., 2012*).

*Figure 8 continued on next page*

*Figure 8 continued*

The online version of this article includes the following video(s) for figure 8:

**Figure 8—video 1.** Confocal z-stack of multiplexed wholemount in situ hybridisation (WISH) of tissue markers for stem *ago2-1* (Smp_179320) (green), muscle *paramyosin* (Smp_085540) (cyan), and nerve *complexin* (Smp_050220) (magenta).

https://elifesciences.org/articles/95628/figures#fig8video1

**Figure 8—video 2.** Confocal z-stack of multiplexed wholemount in situ hybridisation (WISH) of tissue markers for parenchyma (Smp_318890) (green), stem *ago2-1* (Smp_179320) (cyan), and tegument *Meg6* (Smp_163710) (magenta).

https://elifesciences.org/articles/95628/figures#fig8video2

**Figure 8—video 3.** Confocal z-stack of multiplexed wholemount in situ hybridisation (WISH) of tissue markers for tegument *Meg6* (Smp_163710) (green) and protonephridia *ShKT-domain-containing protein* (Smp_335600) (magenta).

https://elifesciences.org/articles/95628/figures#fig8video3

after dissociation by their distinct flask shape but were not identifiable in these 10X data. Perhaps, because of their large size (20 × 50 µm) and irregular shape, they were not incorporated into the 10X GEMs. Therefore, we used plate-based scRNA-seq for the penetration glands to complete the cell-type transcriptome atlas for all known cell types. Together, we have transcriptomically characterised the anatomical systems described by *Pan, 1980*, and we discuss key findings below.

The two muscle clusters in our dataset correspond to the circular and longitudinal muscles that comprise the orthogonal BWMs of the miracidium. We identified differences between circular and longitudinal muscle transcriptomes that have also been observed in planaria; for example *MyoD* and *Wnt-11-1* are expressed in longitudinal and circular muscles, respectively (*Scimone et al., 2017*). Posterior Wnt signalling and anterior Wnt inhibition are commonly observed features of anterior–posterior body patterning in animals (*Petersen and Reddien, 2009*). The expression of *Wnt-11-1* and a Wnt inhibitor, *Notum,* in the circular muscles at opposite poles of the miracidium suggested these muscles are involved in defining the anterior–posterior axis. Furthermore, the unilateral, asymmetrical expression of *Notum* at the anterior end is suggestive of a dorso–ventral axis and/or a left–right axis, neither of which have been identified in the miracidium before. Many axial patterning genes expressed in planaria muscles are essential for patterning during regeneration (*Witchley et al., 2013*; *Vogg et al., 2014*). The expression of orthologues in *S. mansoni* muscles, a nonregenerative animal, suggests that these genes may regulate stem cells during homeostasis in the adult (*Wendt et al., 2020*). As the body axes of the miracidium were defined during embryogenesis, the expression of these genes in the muscles of the larva suggests they will provide positional information to the stem cells of the mother sporocyst in the axial patterning of the daughter sporocysts.

The nervous system accounts for over 50% of cells that make up the larva and is the most transcriptionally heterogeneous tissue, as observed in other animals (*Siebert et al., 2019*; *Vergara et al., 2021*). At the end of embryogenesis, at hatching, most of the miracidia neurons are fully differentiated and form the functional neural circuits of the larva. This is evident in our scRNA-seq data by the presence of discrete, well-defined neural cell clusters. In situ validation of marker genes revealed many of the nerves identified ultrastructurally; for example, the location and shape of Neuron 4 cells resemble that of the bipolar ganglion cells with multi-ciliated nerve endings described by *Pan, 1980*. Examination of their transcriptional profiles indicates their function; top markers include a putative *mTORC1*, two *Polycystin* genes, a *GTP-binding ras-like* and *fibrocystin*. Mammalian orthologues of these function in cells with primary cilia in the detection of flow sheer and regulate mTORC1 signalling to maintain the cell in a quiescent and functional state (*Lai and Jiang, 2020*). Polycystins 1 and 2 expressed in ciliated neurons of the larva of the annelid *Platynereis dumerilli* are involved in mechanosensation, detection of water vibrations and mediate the startle response (*Bezares-Calderón et al., 2018*). Furthermore, in planaria, SoxB1-2 regulates polycystin genes and together they function in ciliated cells that are involved in water flow sensation (*Ross et al., 2018*). The gene expression profile of Neuron 4 cells in the miracidium suggests they could function as flow and/or vibration sensors and may be specified by a Sox transcription factor. These multi-ciliated receptors degenerate 12 hr after the miracidia transform into a mother sporocyst (*Poteaux et al., 2023*).

In situ validation of marker genes also revealed an undescribed cell type, namely the six submuscular cells that express the top marker genes for the ciliary plates (i.e. a 16 kDa CaBP and *p25 alpha*). The expression of these genes in both the ciliary plates and the submuscular cells suggests that these latter cells are transcriptionally similar and might play a role in the development and functioning

of the ciliary plates. Their projection into the external environment suggests a sensory role, which, together with their location between the first and second tier of ciliary plates, could suggest a role in coordinating the movement of the motile cilia across the plates and tiers in response to sensory cues (i.e. similar in function to the ciliomotor pacemaker neurons in *Platynereis* larvae, *Verasztó et al., 2017*). The ciliary plates are a miracidium-specific cell type (it is the only life cycle stage that uses motile cilia for locomotion), and once inside the snail, these plates are shed within 2–48 hr (*Meuleman et al., 1978*; *Poteaux et al., 2023*). As expected, they are transcriptomically typified by enrichment with dynein and high metabolic activity. The 16 kDa CaBP (Smp_096390) is clearly important to the function of ciliary plates as it is highly expressed in data collected from both droplet- and plate-based scRNA-seq platforms. Accordingly, bulk RNA-seq data show this gene was previously detected in the late embryonic stage, miracidia, and early mother sporocyst stage (*Lu et al., 2021*; *Buddenborg et al., 2023*).

Between all the ciliary plates sit the ridges of the tegumental cells. The ridges connect to the cell bodies beneath the body wall musculature, and their transcriptomes show a predicted network of genes involved in protein folding, export, and translocation. This is consistent with ultrastructural studies that have observed abundant RER, ribosomes, Golgi apparatus, and complex carbohydrates in these cell bodies, suggesting they are active in producing exportable proteins (*Meuleman et al., 1978*; *Pan, 1980*). Large numbers of membrane-bound vesicles were also observed and thought to become part of the new tegument membrane (syncytium) that rapidly forms around the mother sporocyst during the early stages of transformation, protecting the mother sporocyst from attack by the snails' amoebocytes (*Pan, 1980*). The RNA velocity analysis has revealed dynamic genes that may be crucial in the maturation of the tegument cells across this life cycle transition, and in turn, for evasion of host defences and parasite survival. This analysis also suggests a developmental trajectory for the tegument cells from the miracidia Delta/Phi stem cells and follows their progeny to terminally differentiated tegument cells in the miracidium and 5-day-old sporocyst. Many of the molecular regulators and markers of adult tegument biogenesis (*Collins et al., 2016*; *Wendt et al., 2018*; *Wendt et al., 2022*) are expressed in the miracidia and sporocyst stem and tegument cells, indicating that there is a conserved molecular programme for the development of the syncytial tegument in the mother sporocyst and its maintenance in the adult worm.

The transcriptomes of the two stem populations in the miracidium suggest they are in different activation states. This finding aligns with the discovery of two stem cell populations in the mother sporocyst that have different proliferation kinetics (*Wang et al., 2013*). The pre-activated state of Delta/Phi stem cells in the miracidia may signify that they are the first population to proliferate once the miracidium transforms into the mother sporocyst inside the snail. These two stem populations may also have different potency states; the co-expression of multiple transcription factors in Delta/Phi, including those specific to the tegument lineage (*Wendt et al., 2018*), indicates a multipotent progenitor stage. Whereas in the Kappa stem cells, the expression of genes that define the germline in the intra-mammalian stages (*Wang et al., 2018*) suggest pluripotency. Supporting this, the RNA velocity analysis shows that *p53-2*, the genome protection gene (*Wendt et al., 2022*), is most highly expressed, and being actively transcribed, in the Kappa population. The incorporation of miracidia single-cell data into existing stem cell scenarios on *S. mansoni* development (*Wang et al., 2013*; *Wang et al., 2018*; *Li et al., 2021*) shows the continuum of the Kappa population from hatching through the intra-molluscan stages to the adult stage inside the mammalian host (*Figure 7*). The finding that *eled*, and many of the germline-specific regulatory programme components, are expressed in some miracidia Kappa cells suggests that when a miracidium hatches, the cells that will contribute to the germline may already be segregated and the molecular regulatory programme that differentiates somatic (Delta/Phi) and germ cell (Kappa) lineages is present. Intriguingly, the presence of spliced-only copies of the germline defining genes *eled* and *boule* could suggest that they are maternal transcripts that have been restricted to the primordial germ cells during embryogenesis, as is the case in Zebrafish embryos (*Fishman et al., 2023*). An alternative explanation is that unspliced transcripts exist for these lowly expressed genes but their abundance was below our threshold for detection.

An unexpected finding regarding the stem cells was the further clustering of the two populations by sex. This is the first identification of any sex differences in the miracidium. This degree of transcriptomic sexual difference was most evident in the Delta/Phi and Kappa stem cell clusters, with no distinct clustering by sex observed in other cell clusters. This could be explained by incomplete gene dosage

compensation in the miracidium. Following investigations of differences in gene dosage compensation across developmental stages using bulk RNA-seq (*Picard et al., 2019*), our study suggests tissue-specific variation could add a further dimension to the role of dosage compensation in schistosomes. A similar phenomenon has been described in mouse embryonic stem cells and in *Drosophila*, where a precise regulation of a gene-by-gene dosage compensation mechanism is critical to the proper development of specific tissues and organs, such as the wing and eye (*Valsecchi et al., 2018*). In contrast to the miracidium, sex differences in undifferentiated stem cells have not been reported in later stages of the schistosome life cycle (*Wang et al., 2018*; *Diaz Soria et al., 2020*; *Wendt et al., 2020*). In fact, it may not have been possible to detect these differences in earlier studies because scRNA-seq data were mapped onto older versions of the genome, where the sex chromosomes were not as well defined (*Buddenborg et al., 2021*). In addition to using an improved reference annotation, the present study has also benefited from a high level of coverage, enabling differences in stem cells to be more easily discerned. Whether these sex-linked gene expression and dosage differences have functional implications in developing male and female sporocysts remains to be investigated.

The current understanding of schistosome development is that differentiated somatic cells of the miracidium do not continue into the daughter sporocyst stage, the cercariae and beyond; it is the stem cells that persist and differentiate into the cell types of the subsequent life stages (*Wang et al., 2018*). In this respect, all miracidium somatic cells are miracidium-only cells. Our atlas shows that their transcriptional signatures are similar to comparable tissue types at later stages (*Diaz Soria et al., 2020*; *Wendt et al., 2020*; *Diaz Soria et al., 2024*) and that there may be conserved tissue-specific molecular developmental programmes that are deployed multiple times during the complex life cycle. Through this atlas, we have shown that even in this simple and sexually monomorphic short-lived stage, there are different populations of muscle, a diverse and distinct range of neural cell types, and that stem cell transcriptomes are already different based on the sex of the miracidium. The molecular definitions and spatial analyses of the miracidia cell types we have presented here allow us to explore the molecular processes and pathways to reveal cell functions and interactions. In addition, high coverage data from individual cells have allowed us to examine how gene expression changes across and within tissue types. Already, a picture is emerging of how the peripheral neurons may detect environmental cues, connect to the brain, muscles and ciliary plates to orientate in the water column; how the parenchymal cells supply nutrients to the surrounding cells; what the contents of the penetration glands may be; and how the tegumental and stem cells are primed for the transition to the intra-molluscan stage.

## Methods

**Key resources table**

| Reagent type (species) or resource | Designation | Source or reference | Identifiers | Additional information |
|---|---|---|---|---|
| Strain, strain background (*Schistosoma mansoni*) | NMRI strain | Wellcome Sanger Institute | | |
| Sequence-based reagent | Probes used for in situ hybridisation | Molecular Instruments | Lot numbers in *Supplementary file 1r* | |
| Commercial assay or kit | Chromium Next GEM Single Cell 3' Reagent Kits v3.1, 10X Genomics 4 rxns | 10X Genomics | Cat# PN-1000128 | |
| Chemical compound, drug | Native *Bacillus licheniformis* Protease | Creative Enzymes | Cat# NATE-0633 | |
| Chemical compound, drug | Liberase TM (Thermolysin Medium) | Roche | Cat# LIBTM-RO | |
| Commercial assay or kit | NEB Next Cell Lysis Buffer | NEB | E6428 | |
| Commercial assay or kit | NEB low input RNA kit | NEB | E6420 | |
| Commercial assay or kit | NEBNext Ultra II FS DNA Library Prep Kit for Illumina | NEB | E7805 | |
| Software, algorithm | ImageJ (Fiji) Version 2.1.0 | Fiji | https://imagej.net/Fiji | |

*Continued on next page*

*Continued*

| Reagent type (species) or resource | Designation | Source or reference | Identifiers | Additional information |
|---|---|---|---|---|
| Software, algorithm | R 3.6.2 | The R foundation | https://www.r-project.org/ | |
| Software, algorithm | RStudio 1.3.1056 | Rstudio | https://rstudio.com | |
| Software, algorithm | Python 3.8.5 | Python | https://www.python.org/downloads/ | |
| Software, algorithm | Cell Ranger 6.0.1 | Cell Ranger | https://www.10xgenomics.com/support/software/cell-ranger/latest | |
| Software, algorithm | Seurat 4.3.0 | Seurat | https://satijalab.org/seurat/articles/install.html | |
| Software, algorithm | kallisto bustools 0.27.3 | kallisto bustools | https://www.kallistobus.tools/ | |
| Software, algorithm | Scanpy 1.8.2 | Scanpy | https://scanpy.readthedocs.io/en/stable/ | |
| Software, algorithm | scvelo 0.2.4 | scvelo | https://scvelo.readthedocs.io/en/stable/ | |
| Software, algorithm | kallisto 0.44.0 | kallisto; *Melsted et al., 2021* | https://pachterlab.github.io/kallisto/ | |
| Software, algorithm | SAM 1.0.1 | SAM; *Tarashansky et al., 2019*; *Zanini and Tarashansky, 2022* | https://github.com/atarashansky/self-assembling-manifold | |
| Other | WormBase ParaSite Release 18 | WormBase ParaSite | https://parasite.wormbase.org/Schistosoma_mansoni_prjea36577/Info/Index/ | Genome version 10 |
| Software, algorithm | Adobe Creative Cloud | Adobe | https://www.adobe.com | |

## Parasite material

*S. mansoni* (NMRI strain) eggs were isolated from livers collected from experimentally infected mice perfused 40 days post-infection. All animal-regulated procedures, including the experimental mouse infections at the Wellcome Sanger Institute, were conducted under Home Office Project Licence No. P77E8A062 held by GR. All the protocols were presented and approved by the Animal Welfare and Ethical Review Body (AWERB) of the Wellcome Sanger Institute. The AWERB is constituted as required by the UK Animals (Scientific Procedures) Act 1986 Amendment Regulations 2012.

## Nuclei count

*S. mansoni* eggs isolated from infected mouse livers were hatched under light in freshwater (*Mann et al., 2010*), and the miracidia collected within 4 hr of hatching. The sample was put on ice for an hour to sink and concentrate the larvae, and all but 1 ml of the water was removed. One ml of 8% paraformaldehyde (4% final PFA concentration) was added to fix the miracidia for 30 min at room temperature. The larvae were rinsed in phosphate-buffered saline (PBS)/Tween (0.1%) and mounted in DAPI Fluoromount for confocal imaging. The whole mounts were imaged on a Leica Sp8 using a ×40 objective with the pinhole set to 1 AU, and a 30–50% overlap between Z slices. The z-stacks were imported into ImageJ, and each nucleus was manually segmented and counted using the plugin TrakEM2 (*Cardona et al., 2012*).

## Single-cell tissue dissociation

*S. mansoni* miracidia for single-cell dissociation were collected as described above. Eggs isolated from mouse livers were hatched in nuclease-free water (NFW) and collected within 4 hr of hatching. Two samples of miracidia were collected; sample 1 contained approximately ~8250 individuals, and sample 2–11,700. The samples (live miracidia in NFW in Low Protein Binding 15 ml Falcon Tubes [cat# 30122216]) were placed on ice for an hour to sink and concentrate the larvae, and all but 20 µl of water was then removed.

Three hundred µl of digestion buffer (2.5 µl of 0.5 M Ethylenediaminetetraacetic acid (EDTA) in 300 µl of Hank's Buffered Salt Solution without $Ca^{2+}$ and $Mg^{2+}$ [HBSS−/−]) was added to the larvae.

We reconstituted 250 mg of the enzyme Native *Bacillus licheniformis* Protease (NATE-0633, Creative Enzymes) (*Adam et al., 2017*) in 2.5 ml of HBSS−/− (to give a stock concentration of 100 mg/ml) and added 200 µl of this stock to the larvae in buffer (final enzyme concentration = 40 mg/ml). The samples were kept on ice for 45 min and pipetted gently ~20 times every 5 min. Two hundred µl of 25% bovine serum albumin (BSA) in 1× HBSS−/− was added to the sample to quench the enzyme, and 10 ml of ice-cold PBS+/+ was added to bring the BSA to ~0.5%. The samples were centrifuged at 350 × *g* for 5 min at 4°C, and all but 20 µl of the supernatant was removed. We reconstituted 5 mg of the enzyme Liberase (Thermolysin Medium) Research Grade (LIBTM-RO Roche) in 2 ml of PBS (to give a stock concentration of 2.5 mg/ml) and added 100 µl of this stock to the sample (final enzyme concentration = 2.08 mg/ml). The samples were incubated at room temperature for 15 min, pipetting gently ~10 times every 5 min. The suspension was passed through 60 and 30 µm cell strainers (Falcon) and resuspended in 5 ml of cold 1× PBS supplemented with 0.5% BSA. The samples were centrifuged at 350 × *g* for 10 min at 4°C, and all but 700 µl of the supernatant was removed.

Sample 1 cell suspension was sorted; 200 µl was used for a no-stain control for autofluorescence during cell counting and sorting. The remaining 500 µl of cell suspension was stained with 1 µg/ml of propidium iodide (PI; Sigma P4864) to label dead/dying cells. Sample 1 was sorted into an eppendorf tube using the BD Influx cell sorter (Becton Dickinson, NJ) by enriching for PI negative cells (i.e. live cells). It took ~3 hr from the beginning of the enzymatic digestion to generating single-cell suspensions ready for library preparation on the 10X Genomics Chromium platform. Sample 2 cell suspension was not sorted.

## 10X Genomics library preparation and sequencing

The 10X Genomics protocol ('Chromium Next GEM Single Cell 3′ Reagent Kits v3.1 User Guide' available from https://assets.ctfassets.net/an68im79xiti/1eX2FPdpeCgnCJtw4fj9Hx/7cb84edaa9eca04b607f9193162994de/CG000204_ChromiumNextGEMSingleCell3_v3.1_Rev_D.pdf) was followed to create gel in emulsion beads (GEMs) containing single cells, hydrogel beads and reagents for reverse transcription, perform barcoded cDNA synthesis, and produce sequencing libraries from pooled cDNAs. The concentration of single-cell suspensions was approximately 119 cells/µl (sample 1) and 122 cells/µl (sample 2), as estimated by flow cytometry-based counting (Beckman Coulter Cytoflex S). We concentrated both samples to 732 cells/µl. We loaded two aliquots of sample 1 and two of sample 2 into four 10X reactions according to the 10X protocol (Chromium Single Cell 3′ Reagent Kits v3.1), intending to capture approximately 10,000 cells per reaction.

Library construction (following GEM breakage) using 10X reagents following the 'Single Cell 3′ Reagent Kits v3.1 User Guide'. The four libraries were sequenced on one lane of Illumina Novaseq XP (28 bp read 1, 90 bp read 2, 10 bp index 1, 10 bp index 2). All raw sequence data are deposited in the ENA under study accession PRJEB45615.

## Mapping and quantification of single-cell RNA-seq

We mapped the single-cell RNA-seq data to version 10 of the *S. mansoni* reference genome (*Buddenborg et al., 2021* and WormBase ParaSite *Howe et al., 2016*; *Howe et al., 2017*) using 10X Genomics Cell Ranger (version 6.0.1) (*Zheng et al., 2017*). Before mapping, exon 8 was removed from Smp_323380 and the 3′ end was trimmed from Smp_337410 to reduce erroneous mapping to the W chromosome from Z gametologs. The Cell Ranger implementation of EmptyDrops (*Lun et al., 2019*) was used to detect cells and empty droplets. Across the four samples, on average 63.88% of the reads mapped confidently to the transcriptome (63.4% for sorted, 64.35% for unsorted), and an average 83.65% of reads were associated with a cell. There were an average of 118,395 reads per cell. Across the four samples, we captured an estimated 33,319 cells: 8815 and 24,504 cells from sorted and unsorted samples, respectively. The median gene count was 1660, and the median UMI count was 5343. Following QC, 20,478 cells remained for onward analysis.

## QC and clustering using Seurat

The four samples were imported into Seurat (version 4.3.0) (*Hao et al., 2021*) using R (version 4.1.3) in RStudio (version 1.4.1106) for QC, clustering, and marker gene identification. The four samples were initially processed separately. Cells with less than 200 genes expressed and genes in less than three cells were removed. QC metrics, including gene distribution and UMI counts

per cell, were inspected visually and used to select QC cut-offs. Following a preliminary analysis without further cell filtering, cells with fewer than 500 UMIs were removed, and a maximum mitochondrial threshold of 5% was applied. This was based on data distribution, best practices from the authors of the software, and other similar studies; here, we were aiming to remove low-quality cells while being relatively permissive as there may be cell types in these data outside of standard filtering parameters. Based on existing knowledge of this larva, we were expecting both multinucleate and anucleate cell type(s) as well as single-nuclei cells (*Pan, 1980*). The data were normalised, scaled (regressing for mitochondrial gene percentage), and variable genes were identified for clustering following Seurat guidelines. Throughout the analysis of these data, several tools were used to select the optimal number of PCs to use: Seurat's ElbowPlot and JackStraw, and molecular cross-validation (https://github.com/constantAmateur/MCVR/blob/master/code.R; *Young, 2022*), and Seurat's clustree tool was used to assess the stability of clusters with increasing resolution. We employed DoubletFinder (*McGinnis et al., 2019*) to identify multiplets, which can lead to spurious signals. The estimated doublet rate was calculated based on 10X loading guidelines and adjusted for our sample concentrations; we estimated multiplet rates at 8% for sorted samples and 52.8% for unsorted samples. We used paramSweep, as recommended, to identify the optimum parameters for each sample in our data and removed cells classified as multiplets. DoubletFinder removed 299 and 303 cells from the two sorted samples, and 6069 and 5496 cells from the two unsorted samples.

We then combined the two sorted samples and the two unsorted samples (producing one sorted object and one unsorted object). We used SCTransform to normalise and scale these two objects (with regression for mitochondrial percentage and sample ID). Seurat's IntegrateData function was then used to integrate the sorted and unsorted samples (with normalization.method = "SCT"). This resulted in 20,478 cells passing all the QC filters and going onto the primary analysis, 3849 and 3962 from the sorted samples and 6702 and 5965 from the unsorted samples.

Following integration, principal component analysis (PCA) was run for up to 100 PCs. The data were explored using a range of PCs and resolutions and using the tools listed above, 55 PCs and a resolution of 1 were selected for the analysis. We manually segmented out the cells in the Neuron 3 cluster based on clear differences in gene expression. Top markers were identified using Seurat's FindAllMarkers (min.pct = 0.0, logfc.threshold = 0.0, test.use = "roc") as recommended by Seurat best practices (https://satijalab.org/seurat/) to capture subtle differences in gene expression, and were sorted by AUC. We used the identity and expression of top marker genes (as identified by AUC), along with previously published cell- and tissue-type marker gene expression, to determine the identity of the cell clusters. To link gene description and chromosome locations to gene IDs, the GFF for the latest publicly available genome version was downloaded from WBPS (WBPS18 release, *Howe et al., 2016*; *Howe et al., 2017*).

## Sex-effect analysis

To investigate sex effects on the single-cell transcriptome, the cells were categorised as W+ or W− (an approximation of female and male cells, respectively, as males lack WSR genes). As ground-truth data were absent, the % of reads mapping to ZSR and WSR was calculated using Seurat's PercentageFeatureSet. Cells with 0.2% or more reads mapping to the WSR were classified as W+, and the remaining cells were classified as W−. The 0.2% threshold was selected (*Figure 5—figure supplement 4*) to avoid classifying ZZ cells with a few transcripts erroneously mapping to a gene on the WSR as W+. This produced 8702 cells labelled as W+, and 11,776 cells labelled as W−.

## SAM algorithm

The count data from the QC-filtered Seurat analysis were exported for SAM (v1.0.1) analysis. For each SAM analysis, the raw count data were imported as a SAM object, standard preprocessing done (using default SAM parameters), and the SAM algorithm (*Tarashansky et al., 2019*) run. For further analysis, the object was converted into a Scanpy (v 1.8.2) (*Wolf et al., 2018*) object, top markers extracted, and metadata from the Seurat analysis were attached. This method was repeated on the Neuron 1 cluster alone, to better understand the transcriptome heterogeneity.

## RNA velocity

All four miracidia samples and the raw data from the mother sporocyst (*Diaz Soria et al., 2024*) were pseudo-mapped using kallisto bustools (v 0.27.3) *Melsted et al., 2021* using the lamanno workflow to the version 10 *S. mansoni* genome (WBPS18 release *Howe et al., 2016*; *Howe et al., 2017*, with the same amendments as listed above). The filtered spliced and unspliced matrices for each sample individually were imported to R. DropletUtils (v 1.20.0) (*Lun et al., 2019*) and BUSpaRse (v 1.14.1) (*Moses, 2023*) were used to process and QC the matrices, and filter out cells and genes with few reads. The matrices were then converted to Seurat objects and further filtered so only cells with the cell barcodes that passed QC filtering in the primary Seurat analysis were retained. The metadata and raw counts of the unspliced and spliced reads following filtering were exported as CSV files. The same method was performed on the mother sporocyst samples, which were combined before exporting. For the joint analyses of miracidia and sporocyst cells, the combined sporocyst Seurat object was merged with each of the four miracidia samples separately (due to the mismatch in sample size and batch effect between sorted and unsorted samples) and all were exported.

The CSV files containing spliced and unspliced reads, together with sample metadata were combined to make an AnnData object using Scanpy (v 1.8.2, Python v 3.8.5) (*Wolf et al., 2018*). Using scvelo (v 0.2.4) (*Bergen et al., 2020*), the standard parameters were used to filter and normalise the data, compute moments, and recover dynamics. Velocity was calculated using the dynamical mode. Following PCA, a UMAP was generated (n_neighbors = 10, n_pcs = 40) and Leiden clustering was added (resolution = 0.8). The top-ranked dynamical genes for each cluster and known genes of interest were plotted. This was performed for each of the four miracidia samples combined with the sporocyst samples, with and without the use of combat for batch correction. As the sporocyst samples were also sorted, we used the sorted miracidia samples to reduce technical effects.

To focus on the stem/germinal and tegument clusters, the same method as above was used after subsetting the data in Seurat to contain only cells from the clusters Stem/germinal, Tegument 1, and Tegument 2 in the sporocyst samples, and only Stem A–G and Tegument clusters from the miracidia clusters.

We present the stem and tegument cells from the sporocyst stage merged with sample 1 (sorted). The equivalent results from sporocysts merged with sample 4 of the miracidia and results with all cells from those two conditions are available in the supplementary data (*Figure 6—figure supplements 4–6*).

## EdU-staining experiment

Miracidia were collected as described above for 2 hr. As miracidia continually hatched over this time period, they ranged from 0 to 2 hr post-hatching. The miracidia were collected and split into two samples: one for EdU labelling and the second for a negative control (miracidia in water without EdU). Sample one was pulsed with 500 μM EdU in water for 4 hr. Sample two was kept in water for 4 hr. Both samples were fixed in 4% PFA for 30 min at room temperature. EdU incorporation was detected by click reaction with 25 μM Alexa Fluor488 Azide conjugates (Invitrogen) for 30 min. As a positive control, we pulsed freshly transformed mother sporocysts in EdU for 3 days (*Wang et al., 2013*). Three biological replicates were carried out, with 20 miracidia and 20 mother sporocysts per treatment examined for each replicate.

## Plate-based scRNAseq analysis of individually isolated cells

A sample of thousands of miracidia was dissociated into a single-cell suspension using the dissociation protocol described above. Individual gland cells, ciliary plates, and unknown cells were picked up one at time using an EZ Grip transfer pipette set to aspirate a 3-μl volume (Research Instruments Ltd 7-72-2800). Each cell was placed in a well of a 96-well plate containing a lysis buffer (NEBNext Cell Lysis Buffer, E6428). cDNA was generated using the NEB low input RNA kit (E6420). DNA library construction was carried out using the NEBNext Ultra II FS DNA Library Prep Kit for Illumina (E7805). The libraries were sequenced on one lane of NovaSeq SP. The reads from the 96 samples were pseudoaligned to the version 10 *S. mansoni* genome (WBPS18 release *Howe et al., 2016*; *Howe et al., 2017*, with the same amendments as listed above using kallisto (v 0.44.0) *Bray et al., 2016*). These were imported into R (v4.3.1) and the metadata were inspected, after which only samples coming from a single cell with a minimum of 20% reads pseudoaligned were retained (*n* = 76).

The samples were then processed using sleuth (v0.30.1) *Pimentel et al., 2017* following guidance from sleuth tutorials, including the steps for normalising, fitting error models, and differential analysis. Differential analysis was used to identify significant genes between the cell types, which were visualised, along with known genes of interest, using sleuth's heatmap function. Furthermore, the kallisto outputs were used to calculate the top 200 genes by median TPM of the two key cell types of interest, ciliary plates and gland cells, and these were used for GO term analysis following the method described below.

### In situ hybridisation

Miracidia collected as described above within 4 hr of hatching were fixed in 4% PFA in 2 µm filtered PBSTw (1× PBS + 0.1% Tween) at room temperature for 30 min on a rocker. The larvae were transferred to incubation baskets (Intavis, 35 µm mesh) in a 24-well plate, rinsed 3 × 5 min in PBSTw, and incubated in 5 µg/ml Proteinase K (Invitrogen) in 1× PBSTw for 5 min at room temperature. They were post-fixed in 4% Formaldehyde in PBSTx for 10 min at room temperature and then rinsed in PBSTw for 10 min. From this point on, the in situ hybridisation experiments followed the protocol described by *Choi et al., 2016* and developed for wholemount nematode larvae. We carried out fluorescent in situ hybridisation on 27 cell-type marker genes. Probes, buffers, and hairpins for third-generation in situ hybridisation chain reaction experiments were purchased from Molecular Instruments (Los Angeles, California, USA) (*Choi et al., 2018*). Molecular Instruments designed probes against the sequences on WormBase ParaSite (WBPS version 18, WS271, *Howe et al., 2017*; *Supplementary file 1R*). We subsequently labelled some larvae with phalloidin by adding 1 µl phalloidin to 1 ml 5× SSCT (Sodium Chloride and Sodium Citrate buffer [Invitrogen AM9763] + 0.1% Tween) (for an hour at room temperature in the dark, then rinsing 6 × 30 min in SSCT, incubating overnight at 4°C in DAPI fluoromount-G (Southern Biotech) and then mounting and imaging on a confocal laser microscope (Leica Sp8)). We carried out at more than three in situ hybridisation experiments for each marker gene we validated (each experiment was a biological replicate). From each experiment, we imaged (by confocal microscopy) 10 miracidia (technical replicates) per marker gene experiment. To count the number of cells belonging to each tissue type we imported confocal z-stacks of tissue marker gene expression (*Figure 8—videos 1–3*) into ImageJ, and using the plugin TrakEM2 (*Cardona et al., 2012*), manually segmented each nucleus that was expressing, and was surrounded, by the transcripts.

### Opsin phylogeny

Amino acid sequences for selected metazoan opsin proteins were obtained from the GenBank database and aligned using the – auto mode of MAFFT v7.450 (*Katoh and Standley, 2013*), then trimmed using trimAl v1.4.rev22 (*Capella-Gutiérrez et al., 2009*) to remove alignment columns with more than 50% gap characters ensuring at least 50% of alignment columns were maintained in the final alignment. Phylogenetic analysis of this alignment was performed using RaxML v8.2.12 (*Stamatakis, 2014*). The best-fitting empirical amino acid substitution model was identified as maximising the log-likelihood of the data under a model incorporating gamma-distributed variation in substitution rates across sites, and this optimal model (Le Gascuel,LG) was used for subsequent inference. The maximum-likelihood tree and branch lengths were identified as the highest likelihood from 25 independent searches starting from different random starting tree topologies, and support for partitions on this tree was estimated using 1000 bootstrap replicates of resampling alignment positions with replacement, using the fast bootstrap (−x) algorithm implemented in RaxML.

### Protein–protein interaction analysis

Protein interactions were predicted using the online search tool STRING https://www.string-db.org/; V 11; *Szklarczyk et al., 2019* for the following well-defined cell clusters: Neurons 3–5, ciliary plates, protonephridia, Parenchyma 1 and 2, tegument, and the two main stem cell populations, Delta/Phi and Kappa. Protein sequences for the top 100 marker genes for each cell cluster were collected from WormBase ParaSite. The protein sequences were entered as a multiple protein search. Default settings were used to predict interactions with either a minimum interaction (confidence) score of 0.4 or 0.7, corresponding to a medium or high level of confidence.

## GO enrichment analysis

Each cluster's list of marker genes was filtered to focus on specific cluster markers. Genes were retained for analysis if they had a minimum AUC score of 0.7. To compare the functional enrichment in Stem 1 vs Stem 2, Stem A was combined with Stem B and Stem C was combined with Stem D, and the differentially expressed markers were identified using Seurat's FindMarkers. These markers were filtered (with adjusted p-values <0.001) and used as the basis for GO enrichment analysis in the Delta/Phi and Kappa stem clusters. TopGO (version 2.46.0) (*Alexa and Rahnenfuhrer, 2023*) was used in R with the weight01 method to identify enriched GO terms. Node size was set to a minimum of five terms, and analysis was run for each cluster on the Biological Process and Molecular Function categories. Fisher's exact test was used to assess the statistical significance of overrepresented terms, with an FDR threshold set <0.05. The same method was used to identify overrepresented terms in the WSR genes, ZSR genes, and WSR together with ZSR genes for the BP, MF, and Cellular Component categories.

## Acknowledgements

We thank Simon Clare and his team for technical assistance with the life cycle maintenance and collection of parasite material at the Wellcome Sanger Institute, and Karl Hoffmann, Josephine Fforde-Thomas and Benjamin Hulme for additional ongoing parasite material. We thank Sarah Buddenborg for work on the annotation of the *Schistosoma mansoni* genome; David Goulding for assistance with Microscopy; teams within DNA Pipelines for data production; and core Informatics teams and Pathogen Informatics for the infrastructure used for data analysis. We thank members of the Parasite Genomics team at the Wellcome Sanger Institute for their comments and input on this study. We thank Alan Wilson and Uriel Koziol for comments on the manuscript. Finally, we thank the two reviewers and the editor for their valuable feedback on this manuscript. The project was funded by the Wellcome Trust (grant 098051 and 206194). KR also received transition funds from the Marine Biological Laboratory in Woods Hole. SRD and GR are supported by UKRI Future Leaders Fellowships [MR/T020733/1] and [MR/W013568/1], respectively.

## Additional information

### Funding

| Funder | Grant reference number | Author |
|---|---|---|
| Wellcome Trust | 10.35802/098051 | Matthew Berriman |
| Wellcome Trust | 10.35802/206194 | Matthew Berriman |
| Marine Biological Laboratory | | Kate A Rawlinson |
| UK Research and Innovation | MR/T020733/1 | Stephen R Doyle |
| UK Research and Innovation | MR/W013568/1 | Gabriel Rinaldi |

The funders had no role in study design, data collection, and interpretation, or the decision to submit the work for publication. For the purpose of Open Access, the authors have applied a CC BY public copyright license to any Author Accepted Manuscript version arising from this submission.

### Author contributions

Teresa Attenborough, Conceptualization, Data curation, Formal analysis, Investigation, Methodology, Project administration, Writing – original draft, Writing – review and editing; Kate A Rawlinson, Formal analysis, Data curation, Funding acquisition, Validation, Investigation, Visualization, Methodology, Writing – original draft, Project administration, Writing – review and editing; Carmen L Diaz Soria, Kirsty Ambridge, Jennie Graham, Methodology; Geetha Sankaranarayanan, Investigation, Methodology; James A Cotton, Data curation; Stephen R Doyle, Investigation; Gabriel Rinaldi, Formal

analysis, Funding acquisition, Investigation, Methodology, Writing – review and editing; Matthew Berriman, Supervision, Funding acquisition, Project administration, Writing – review and editing

## Author ORCIDs
Kate A Rawlinson ⓘ https://orcid.org/0000-0001-8297-8405
Stephen R Doyle ⓘ https://orcid.org/0000-0001-9167-7532
Matthew Berriman ⓘ http://orcid.org/0000-0002-9581-0377

## Ethics
All animal-regulated procedures, including the experimental mouse infections at the Wellcome Sanger Institute, were conducted under Home Office Project Licence No. P77E8A062 held by Gabriel Rinaldi. All the protocols were presented and approved by the Animal Welfare and Ethical Review Body (AWERB) of the Wellcome Sanger Institute. The AWERB is constituted as required by the UK Animals (Scientific Procedures) Act 1986 Amendment Regulations 2012.

Reviewer #1 (Public Review): https://doi.org/10.7554/eLife.95628.3.sa1
Reviewer #2 (Public Review): https://doi.org/10.7554/eLife.95628.3.sa2
Author response https://doi.org/10.7554/eLife.95628.3.sa3

# Additional files

## Supplementary files
• Supplementary file 1. All supplementary tables relating to this study. (**a**) Samples and single cell RNA-seq (scRNA-seq) mapping statistics. (**b**) Seurat marker genes in all cell clusters. (**c**) Seurat top 5 marker genes in all cell clusters. (**d**) Contribution of sorted and unsorted samples to the 19-cell clusters. (**e**) Muscle clusters 1 and 2 differential gene expression analysis. (**f**) Marker genes for Neuron 1 self-assembling manifold (SAM) subclusters. (**g**) Gene ontology (GO) enrichment terms. (**h**) Top 200 genes by TPM (transcripts per million) in gland cells from plate-based scRNA-seq. (**i**) GO terms from the top 200 genes by TPM in gland cells and ciliary plates from NEB-seq. (**j**) Significant gene expression differences between gland cells and ciliary plates from plate-based scRNA-seq. (**k**) Significant gene expression differences between gland cells and other cells from plate-based scRNA-seq. (**l**) Top 200 genes by TPM in ciliary plates from plate-based scRNA-seq. (**m**) GO enrichment terms for W-specific region (WSR), Z-specific region (ZSR), and WSR + ZSR genes. (**n**) Stem clusters Delta/Phi and Kappa differential gene expression. (**o**) GO enrichment terms of stem differentially expressed genes. (**p**) Top 5 dynamical genes from each cluster from stem and tegument RNA velocity analysis. (**q**) Comparison of tissue-type composition of the miracidium via in situ hybridisation and the scRNA-seq data. (**r**) In situ hybridisation probes.

• MDAR checklist

## Data availability
All raw sequence data are available at the ENA under study accession PRJEB45615, and the single-cell samples are available under Run Accessions ERR12372709 (miracell10376067), ERR12372706 (miracell10376068), ERR12372707 (miracell10376069), and ERR12372708 (miracell10376070). Plate based sample run accessions range from ERR12372710 to ERR13441833, for individual sample IDs and run accessions see *Supplementary file 1a*. The scripts used to perform the analyses presented here are available at https://github.com/tessatten/singlecell-miracidia, (copy archived at *Attenborough, 2024*).

The following dataset was generated:

| Author(s) | Year | Dataset title | Dataset URL | Database and Identifier |
|---|---|---|---|---|
| Attenborough T, Rawlinson KA, Diaz Soria CL, Ambridge K, Sankaranarayanan G, Graham J, Cotton JA, Doyle SR, Rinaldi G, Berriman M | 2023 | Schistosoma_mansoni_ miracidia_single_cell_ RNAseq | https://www.ebi.ac.uk/ena/browser/view/PRJEB45615 | European Nucleotide Archive, PRJEB45615 |

The following previously published dataset was used:

| Author(s) | Year | Dataset title | Dataset URL | Database and Identifier |
|-----------|------|---------------|-------------|-------------------------|
| Diaz Soria CL, Attenborough T, Lu Z, Fontenla S, Graham J, Hall C, Thompson S, Andrews TGR, Rawlinson KA, Berriman M, Rinaldi G | 2022 | Single_cell_transcriptome_of_schistosome_sporocysts | https://www.ebi.ac.uk/ena/browser/view/PRJEB52467 | European Nucleotide Archive, PRJEB52467 |

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
