## [Editor Report · eLife assessment]

This is a **valuable** study in which the authors provide an expression profile of the human blood fluke, Schistosoma mansoni. A strength of this **solid** study is in its inclusion of in situ hybridisation to validate the predictions of the transcript analysis.

---

## [Referee Report · Reviewer #1 (Public Review)]

In this work, the authors provide a valuable transcriptomic resource for the intermediate free-living transmission stage (miracidium larva) of the blood fluke. The single-cell transcriptome inventory is beautifully supplemented with in situ hybridization, providing spatial information and absolute cell numbers for many of the recovered transcriptomic states. The identification of sex-specific transcriptomic states within the populations of stem cells was particularly unexpected. The work comprises a rich resource to complement the biology of this complex system.

Comments on revised version:

I have read through the responses and the revised manuscript. I think together this results in an improved version.

---

## [Referee Report · Reviewer #2 (Public Review)]

Summary:

In this manuscript the authors have generated a single-cell atlas of the miracidium, the first free-living stage of an important human parasite, Schistosoma mansoni. Miracidia develop from eggs produced in the mammalian (human) host and are released into freshwater, where they can infect the parasite's intermediate snail host to continue the life cycle. This study adds to the growing single-cell resources that have already been generated for other life-cycle stages and, thus, provides a useful resource for the field.

Strengths:

Beyond generating lists of genes that are differentially expressed in different cell types, the authors validated many of the cluster-defining genes using in situ hybridization chain reaction. In addition to providing the field with markers for many of the cell types in the parasite at this stage, the authors use these markers to count the total number of various cell types in the organism. Because the authors realized that their cell isolation protocols were biasing the cell types they were sequencing, they applied a second method to help them recover additional cell types.

Schistosomes have ZW sex chromosomes and the authors make the interesting observation that the stem cells at this stage are already expressing sex (i.e. W)-specific genes.

Comments on revised version:

The manuscript has been improved after revisions. The methods, data and analyses broadly support the claims with only minor weaknesses.

---

## [Author Response]

The following is the authors’ response to the original reviews.

**eLife assessment**
This is a valuable study in which the authors provide an expression profile of the human blood fluke, Schistosoma mansoni. A strength of this solid study is in its inclusion of in situ hybridisation to validate the predictions of the transcript analysis.

We thank the reviewers and the editor for their effort and expertise in reviewing our manuscript. We have made changes based on the reviews and believe this has greatly strengthened our manuscript. We appreciate their insightful comments and suggestions.

**Public Reviews:**

**Reviewer #1 (Public Review):**
In this work, the authors provide a valuable transcriptomic resource for the intermediate free-living transmission stage (miracidium larva) of the blood fluke. The single-cell transcriptome inventory is beautifully supplemented with in situ hybridization, providing spatial information and absolute cell numbers for many of the recovered transcriptomic states. The identification of sex-specific transcriptomic states within the populations of stem cells was particularly unexpected. The work comprises a rich resource to complement the biology of this complex system, however falls short in some technical aspects of the bioinformatic analyses of the generated sequence data.(1) Four sequencing libraries were generated and then merged for analysis, however, the authors fail to document any parameters that would indicate that the clustering does not suffer from any batch effects.

We thank the reviewer for this comment which has given us the opportunity to elaborate on this interesting point. Consequently, we have added evidence to show that the data do not suffer from batch effects between samples (e.g. between sorted samples 1 and 4, and unsorted samples 2 and 3). We now show that there are contributions to all clusters from sorted and unsorted samples and highlight the benefits to using both conditions in a cell atlas with unknown cell types.

Accordingly, we have now added the following paragraph to line 153:

There were contributions from sorted and unsorted samples in almost all clusters (except ciliary plates). We found that some cell/tissue types had similar recovery from both methods (e.g. Stem A, Muscle 2, and Tegument), others were preferentially recovered by sorting (e.g Neuron 1, Neuron 4, and Stem E), and some were depleted by sorting (e.g. Parenchyma 1, Protonephridia, and Ciliary plates) (Supplementary Figure 1 , Supplementary Table 4). This variation in recovery, therefore, enabled us to maximise the discovery and inclusion of different cell types in the atlas.

We have now added a Supplementary Figure 1 showing the contribution of sorted and unsorted cells to the Seurat clusters. We have also included a Supplementary Table 4 detailing the cell number contribution for both conditions and the percentages in order to easily compare differential recovery between cell types.

These are added to the manuscript.

(2) Additionally, the authors switch between analysis platforms without a clear motivation or explanation of what the fundamental differences between these platforms are. While in theory, any biologically robust observation should be recoverable from any permutation of analysis parameters, it has been recently documented that the two popular analysis platforms (Seurat - R and scanPy python) indeed do things slightly differently and can give different results (https://www.biorxiv.org/content/10.1101/2024.04.04.588111v1). For this reason, I don't think that one can claim that Seurat fails to find clusters resolved by SAM without running a similar pipeline on the cluster alone as was done with SAM/scanPy here. The manuscript itself needs to be checked carefully for misleading statements in this regard.

We thank the reviewer for this comment and agree that it’s important to increase the clarity on this matter. We have added additional detail to explain that results of subclustering Neuron 1 using Seurat and SAM/ScanPy were broadly similar, but that we presented the results from the SAM/ScanPy analysis due to the strengths of SAM in detecting small differences in gene expression (Tarashanky et al., 2019 PMID: 31524596). We have included here the UMAP showing subclustering of Neuron 1 in Seurat for comparison.

**Author response image 1. sa3fig1:** UMAP showing subclustering of Neuron 1 cluster in Seurat (SCT normalisation, PC = 19, resolution = 0.3).

We’ve added this additional text to the ‘Neuron abundance and diversity’ section on line 220:

We explored whether Neuron 1 could be further subdivided into transcriptionally distinct cells by subclustering (Supplementary Figure 2; Supplementary Table 6) using the self-assembling manifold (SAM) algorithm (Tarashansky et al., 2019) with ScanPy (Wolf et al., 2018), given its reported strength in discerning subtle variation in gene expression (Tarashansky et al., 2019), although a similar topology was subsequently found using Seurat.

(3) Similarly, the manuscript contains many statements regarding clusters being 'connected to', or forming a 'bridge' on the UMAP projection. One must be very careful about these types of statements, as the relative position of cells on a reduced-dimension cell map can be misleading (see Chari and Pachter 2023). To support these types of interpretations, the authors should provide evidence of gene expression transitions that support connectivity as well as stability estimates of such connections under different parameter conditions. Otherwise, these descriptors hold little value and should be dropped and the transcriptomic states simply defined as clusters with no reference to their positions on the UMAP.

We thank the reviewer for this thoughtful comment. We agree and have rephrased those statements accordingly e.g. line numbers 218, 439, 543, and 557.

(4) The underlying support for the clusters as transcriptomically unique identities is not well supported by the dot plots provided. The authors used very permissive parameters to generate marker lists, which hampers the identification of highly specific marker genes. This permissive approach can allow for extensive lists of upregulated genes for input into STRING/GO analyses, this is less useful for evaluating the robustness of the cluster states. Running the Seurat::FindAllMarkers with more stringent parameters would give a more selective set of genes to display and thereby increase the confidence in the reader as to the validity of profiles selected as being transcriptomically unique.

The Reviewer is correct in noting that we used a permissive approach to enable a better understanding of the biology of each cluster, based on analysing enriched functions. However, we disagree about the suitability of the approach for finding markers. First, the permissive approach produced longer candidate lists, but those with the best AUC scores for each cluster are at the top of the list for each cluster. Second, some of the markers with lower expression also revealed interesting biology (e.g. Notum in the muscles). Furthermore, we used filtering on the marker genes lists to increase the minimum marker gene scores for analyses such as the GO analyses (details in the GO section of the methods). It’s important to stress that our approach also utilised validation by FISH for top marker genes, as well as biologically informative genes that were lower down the marker gene list.

(5) Figure 5B shows a UMAP representation of cell positions with a statement that the clustering disappears. As a visual representation of this phenomenon, the UMAP is a very good tool, however, to make this statement you need to re-cluster your data after the removal of this gene set and demonstrate that the data no longer clusters into A/B and C/D.

We’ve added Supplementary Figure 13 to show that after removing WSR and ZSR genes and reclustering, the data no longer clusters in A/B and C/D, even at a higher resolution where clusters appear oversplit.

Also, as a reader, these data beg the question: which genes are removed here? Is there an over-representation of any specific 'types' of genes that could lead to any hypotheses of the function? Perhaps the STRING/GO analyses of this gene set could be informative.

We have performed GO-enrichment analyses on W-specific genes, Z-specific genes and both together compared to the rest of the genome, but we did not find very informative results (see Supplementary Table 13 that we have now added, line 464). This may be due to the large difference in size. There are approx 900 Z-specific genes (males two copy, females one copy), while approx 30 W-specific genes many of which have homologs in the Z-specific region of the genome. Instead we suggest that tissue-specific regulation of gene dosage compensation is the more likely explanation as reported for other species (Valsecchi et al. 2018).

(6) How do the proportions of cell types characterized via in situ here compare to the relative proportions of clusters obtained? It does not correspond to the percentages of the clusters captured (although this should be quantified in a similar manner in order to make this comparison direct: 10,686/20,478 = ~50% vs. 7%), how do you interpret this discrepancy? While this is mentioned in the discussion, there is no sufficient postulation as to why you have an overabundance of the stem cells compared to their presence in the tissue. While it is true that you could have a negative selection of some cell types, for example as stated the size of the penetration glands exceeds both that of the 10x capabilities (40uM), and the 30uM filters used in the protocol, this does not really address why over half of the captured cells represent 'stem cells'. A more realistic interpretation would be biological rather than merely technical. For example, while the composition of the muscle cells and the number of muscle transcriptomes captured are quite congruent at ~20%, the organism is composed of more than 50% of neurons, but only 15% of the transcriptomic states are assigned to neuronal. Could it be that a large fraction of the stem cells are actually neural progenitors? Are there other large inconsistencies between the cluster sizes and the fraction of expected cells? Could you look specifically at early transcription factors that are found in the neurons (or other cell types) within the various stem cell populations to help further refine the precursor/cell type relationships?

Yes, it is really interesting that more than 50% of cells in the animal are neurons whereas more than 50% of cells in scRNAseq data are stem cells. This dataset provides a unique opportunity to compare tissue composition in the whole animal to the corresponding single cell RNAseq dataset.

The table (in Supplementary Table 17) shows the percentage of cells from each tissue type in the miracidium (identified via in situ hybridisation of tissue-type marker genes) and in the scRNAseq to understand this phenomenon.

This table shows that the single cell protocol used in this study negatively selected for nerves and tegument, and positively selected for stem and parenchyma. The composition of the muscle and protonephridia cells and the number of muscle and protonephridia transcriptomes captured are quite congruent.

This technical finding is also biologically consistent. For instance, the tegument cells span the body wall muscles, with the cell bodies below and a syncytial layer above. It is not known how the tegument fragments during the dissociation process, and which parts of the cells get packaged by the 10X GEMs. Because of tegumental structure, the cells are likely prone to damage, and therefore we speculate that is why the tegument cells are under-represented in our 10X data. Unusually shaped fragments may not have been captured in 10X GEMs and of those that were, damaged or distressed tegument cells/fragments may have been excluded post-sequencing, by QC filters including cell calling, mitochondrial percentage and low transcript count (e.g. if there there was a tegumental fragment with 100 transcripts it would have not passed QC). Stem cells are spherical with a large nucleus:cytoplasm ratio, likely making them more robust during dissociation and more likely to be captured in 10X GEMs.

We don’t think that a large fraction of the stem cells are actually neural progenitors because:

(1) we used previously reported marker genes of different tissue types to identify the single cell RNAseq clusters, e.g. Ago2-1 for stem cells, which has been used in multiple life stages.

(2) The stem cell transcriptomes express many previously reported stem cell marker genes.

(3) We found that the stem cells from the single cell data generally had higher numbers of transcripts than the other cell types which is consistent with the Wang et al. 2013 observation that RNA marker POPO-1 could distinguish germinal (stem) cells from other cell types as they are RNA rich.

(4) We also found higher numbers of ribosomal related transcripts in our stem cell transcriptomes, which is consistent with Pan’s observation that part of the distinct morphology of stem cells is densely packed ribosomes in the cytoplasm.

In order to elaborate on this discussion we have generated new visualisations:

(1) A UMAP of the stem cell marker ago2-1 (Supplementary figure 10), to further illustrate our evidence in classifying the stem cell clusters

(2) A co-expression plot of the stem cell marker ago2-1 with neural marker complexin to confirm that there is little coexpression (the most coexpression being in Neuron 1 and Stem F). We identified that 15.56% of cells in the Stem F cluster show some expression of complexin (neural marker), suggesting that a small fraction of Stem F may be early/precursor neurons, but the gene expression indicates that the majority of cells in Stem F are more likely to be stem cells than any other tissue type. There is little to no complexin expression in the other stem clusters.

(3) Expression plots of the 5 neurogenins (TFs involved in neuronal differentiation) we could identify using WormBase ParaSite in these data. Four of the five showed very little expression, and not in specific clusters. The fifth (Smp_072470) showed slightly more expression, though still sparse, mostly across the stem and neural clusters not enough to indicate that any of the stem clusters are neural progenitors.

**Author response image 2. sa3fig2:** Coexpression UMAP showing the expression of stem cell marker Ago2-1 and neural marker complexin.

**Author response image 3. sa3fig3:** UMAPs showing the expression five putative neurogenins of S. mansoni.

**Reviewer #2 (Public Review):**
Summary:In this manuscript the authors have generated a single-cell atlas of the miracidium, the first free-living stage of an important human parasite, Schistosoma mansoni. Miracidia develop from eggs produced in the mammalian (human) host and are released into freshwater, where they can infect the parasite's intermediate snail host to continue the life cycle. This study adds to the growing single-cell resources that have already been generated for other life-cycle stages and, thus, provides a useful resource for the field.Strengths:Beyond generating lists of genes that are differentially expressed in different cell types, the authors validated many of the cluster-defining genes using in situ hybridization chain reaction. In addition to providing the field with markers for many of the cell types in the parasite at this stage, the authors use these markers to count the total number of various cell types in the organism. Because the authors realized that their cell isolation protocols were biasing the cell types they were sequencing, they applied a second method to help them recover additional cell types.Schistosomes have ZW sex chromosomes and the authors make the interesting observation that the stem cells at this stage are already expressing sex (i.e. W)-specific genes.Weaknesses:The sample sizes upon which the in situ hybridization results and cell counts are based are either not stated (in most cases) or are very small (n=3). This lack of clarity about biological replicates and sample sizes makes it difficult for the reader to assess the robustness of the results and the extremely small sample sizes (when provided) are a missed opportunity to explore the variability of the system, or lack thereof.

We have now added more details about the methods we used for validating cell type marker genes by in situ hybridisation. We have added to the methods that ‘We carried out at least three in situ hybridisation experiments for each marker gene we validated (each experiment was a biological replicate). From each experiment we imaged (by confocal microscopy) at least 10 miracidia (technical replicates) per marker gene experiment.’ on line 1036.

In the figure legends we have added the number of miracidia that were screened, and documented the percentage of the screened larvae that showed the in situ gene expression pattern that is seen in the images in the figures, and that we described in the text.

We manually segmented the nuclei of pan tissue marker genes, and we did this for one miracidium in the case of all tissues, except stem cells where we segmented stem cells in five larvae. Manual segmentation of gene expression in a confocal z-stack is very time consuming. We consider that the variability of different cell and tissue types (stereotypy) between miracidia is beyond the scope of this paper and can be investigated in future work.

Although assigning transcripts to a given cell type is usually straightforward via in situ experiments, the authors fail to consider the potential difficulty of assigning the appropriate nuclei to cells with long cytoplasmic extensions, like neurons. In the absence of multiple markers and a better understanding of the nervous system, it seems likely that the authors have overestimated the number of neurons and misassigned other cell types based on their proximity to neural projections.

This is a valid point, and we acknowledge the difficulties of assigning a nucleus to a cell using mRNA expression only and in the absence of a cell membrane marker. We tried to address this issue by labelling the cell membranes using an antibody against beta catenin after the HCR in situ protocol. This method has been used successfully on sections on slides (Schulte et al., 2024), but we failed to get usable results in our miracidia whole-mounts. The beta catenin localisation marked the membranes of the gland cells but didn’t do the same for the neurons or other cell types (see image below).

**Author response image 4. sa3fig4:** Image showing a maximum intensity projection of a subvolume of a confocal z-stack of a miracidia wholemount in situ hybridisation (by HCR) for paramyosin counterstained with a beta catenin antibody (1:600 concentration of Sigma C2206). The cell membrane of a lateral gland is clearly labelled, but those of the neurons of the brain and the paramyosin+ muscle cells are not.

Our observation that 57% of the cells in a miracidium are nerves is high compared to the *C. elegans* hermaphrodite adult in which 302 out of 959 cells are neurons (Hobert et al., 2016), few studies have equivalent data with which to make comparisons. Despite this, and the limitation described above, we believe that we have not overestimated the number of neural cells. During the process of validating the marker genes and closely examining gene expression in hundreds of miracidia, we noted that the nuclei of different tissue types are distinct and recognisable (see figure below). The nuclei of stem, tegument and parenchymal cells are comparatively large and spherical with obvious nucleoli (i). The four nuclei of the apical gland cell are angular, pentagonal in shape and sit adjoining each other (inside red dashed circle, i-iii), those of the two lateral glands are bilaterally symmetrical and surrounded by flask shaped cytoplasm (arrows, iv). The nuclei of the body wall muscle cells are peripheral and flattened on the outer edge (iii). The notum+ muscle cell nuclei are anterior of the apical gland (manuscript Figure 2E). The only other two tissue types are the nerves and protonephridia, and their nuclei are smaller and more compact/condensed. In situ expression of the protonephridia marker suggests that 6 cells make up the protonephridial system (manuscript Figure 4 B&E). Therefore, by process of elimination, the remaining nuclei should belong to neurons. The complexin expression pattern supports this and we counted 209 nuclei that were surrounded by cpx transcript expression. To help the reader interpret this for themselves we have added confocal z-stacks of miracidia where tissue level markers have been multiplexed (supplementary videos 18-20). We counted all tissue type cells individually and the tissue type cell numbers added up to the overall cell count.

**Author response image 5. sa3fig5:** Image showing the diversity of nucleus morphology between tissue types in the miracidium.

Biologically, it is not surprising that this larva is dominated by neural cells. It must navigate a complex aquatic environment and identify a suitable mollusc host in less than 12 hours. It is a non-feeding vehicle that must deliver the stem cells to a suitable environment where they can develop into the subsequent life cycle stage. Accordingly, the cell type composition reflects this challenge.

The conclusion that germline genes are expressed in the miracidia stem cells seems greatly overstated in the absence of any follow-up validation. The expression scales for genes like eled and boule are more than 3 orders of magnitude smaller than those used for any of the robustly expressed genes presented throughout the paper. These scales are undefined, so it isn't entirely clear what they represent, but neither of these genes is detected at levels remotely high (or statistically significant) enough to survive filters for cluster-defining genes.Given that germ cells often develop early in embryogenesis and arrest the cell cycle until later in development, and that these transcripts reveal no unspliced forms, it seems plausible that the authors are detecting some maternally supplied transcripts that have yet to be completely degraded.

We agree that the expression of genes such as eled and boule are low. We made this clear in the figure legends and text, and have now added scale information to the figure legends. We did not explore these genes as cluster-defining genes, partly due to their comparatively low levels of expression, but as genes already reported to be important in germ line specification. We found the expression of these genes to be consistent with our hypothesis that the Kappa stem cells may include germ line segregated cells, but our hypothesis does not rest on these lower-expressed genes.

It is certainly possible that we have detected some maternally supplied transcripts in the miracidia stem cells. However experiments to distinguish between zygotic and maternal transcripts using metabolic labelling of zygotic transcripts (e.g. Fishman et al. 2023) would be hard in this species due to the hard egg capsule and its ectolethical embryogenesis. Therefore this is out of scope for this work, but this would be a very interesting topic to follow up on and develop tools for.

We have added these sentences to the Discussion ln 746 ‘Intriguingly, the presence of spliced-only copies of the germline defining genes eled and boule could suggest that they are maternal transcripts that have been restricted to the primordial germ cells during embryogenesis, as is the case in Zebrafish embryos (Fishman et al., 2023). An alternative explanation is that unspliced transcripts exist for these lowly expressed genes but their abundance was below our threshold for detection.’

**Reviewer #1 (Recommendations For The Authors):**
Ln 138: specify the version of Seurat used, and reference the primary papers for this software. Also, from the dot plot shown here, these do not all appear to be supported by unique gene sets. How was the final clustering determined? This information is in the methods section, but a summary here could make it more robust for the readership.

In addition to the details in the methods section, we have added the version and referenced the version-specific primary paper for Seurat when it is first mentioned. We have also summarised the methods used to select the final clustering when we first present the results to aid in clarity.

We added to line 140 ‘Using Seurat (version 4.3.0) (Hao et al., 2021), 19 distinct clusters of cells were identified, along with putative marker genes best able to discriminate between the populations (Figure 1C & D and Supplementary Table 2 and 3). We used Seurat’s JackStraw and ElbowPlot, along with molecular cross-validation to select the number of principal components, and Seurat’s clustree to select a resolution where clusters were stable (Hao et al., 2021).’

Ln 147: isn't seven stem cell clusters a lot? See comment in public review.

We did not have preconceived expectations of the number of stem cell clusters, and were guided by the data and gene expression. In doing so we also discovered that four of those clusters were likely only two ‘biologically or functionally distinct’ clusters, but these split into four clusters based on the expression of genes on the sex-specific regions of the chromosomes, which was both unexpected and interesting.

Figure 1D: gene model names are un-informative for the general reader. Can you provide any putative gene identities here to render this plot interpretable? For example in the main text you state that Smp-085540 is paramyosin; please use this annotation in all your visual material (as is used in Figure 2A).

We have added gene names to the dotplots in all figures with the locus identifier (minus the ‘Smp’ prefix) in brackets after the gene name.

Ln 191:196 Identification of the two muscle clusters as circular and longitudinal muscles is very well supported. However, it would be interesting to look specifically at the genes that are different here. Did the authors attempt to specifically pull out genes differentially expressed between these two groups, or only examine the output of FindAllMarkers at this point?

We did indeed look specifically for genes differentially expressed between the muscle clusters, the results of which can be found in Supplementary Table 5 (Line 206). This analysis revealed “Wnt-11-1 (circular) and MyoD (longitudinal) were among the most differentially expressed genes”, which were important findings in our understanding of the muscle cells in the miracidium.

Ln 207: "connected to stem F" - does this refer specifically to their relative positions on the UMAP in Figure 1C? One must be very careful about these types of statements, as the relative position of cells on a reduced-dimension cell map can be misleading (public review).

We agree, and have rephrased accordingly.

Ln 209:211: Here the authors switch from Seurat (R) as an analysis package, to SAM (python) for subset analysis of one large neural cluster. The results indicate that there may be small populations of transcriptomically distinct neural subtypes also within the neural1 cluster, but that the vast majority of these cells do not express unique transcriptomic profiles. Also in the supplementary material for this (SF1) there is a question of whether or not there is any clustering according to batch effects.In general, I find the neuronal section a little difficult to follow and it is unclear how many unique profiles are present and which are documented with in situ. I would recommend re-running the analysis on the entire neural subset (n1:5: complexin positive) and generating an inventory of putatively unique neural states with the associated in situ validation altogether in a main figure.

In response to comments above we have both clarified our reasoning for using SAM analysis, and presented more details on possible batch effects. We have gone through the neural system results in order to make it clearer for the reader to follow.

Ln 236: here the authors introduce a STRING analysis for the first time. Also, this method requires some introduction for the general audience in terms of its goals and general functionality and output.

We used STRING analysis on some well defined clusters to provide additional clues about function. At the first mention of STRING (neuron 3 results) we have added the following statement to give more introduction to the reader: “STRING analysis of the top 100 markers of Neuron 3 predicted two protein interaction networks with functional enrichment: ….”

Ln. 280:281. It is unclear why Steger et al is referenced here. In what way does a description of neural and glandular cell transcriptomic similarity in a Cnidarian inform your data on a member of the playhelmenthes? (which should also be referenced in the introduction: to which phylogenetic lineage does Schistosoma belong).

We have now added that the Schistosoma belong to the Platyhelminths on the first line of the introduction.

Ln 295 we have added ‘We expected to find a discrete cluster(s) for the penetration glands, and that it would show similarities to the neural clusters (as glandular cells arise from neuroglandular precursor cells in other animals, such as the sea anemone, *Nematostella vectensis*, Steger et al., 2022).’

Ln 339: explain the motivation for generating a further plate-based scRNA of the ciliary plates.

We wished to include the ciliary plates alongside the gland cells for plate based RNAseq as they are unique to the miracidium stage and wanted to make sure we had captured them in this study.

Ln 345: Define the tegumental cells for the general reader.

We have added further description on tegument cells in the introduction and tegument results section, e.g. on line 61, 366.

Ln 365: "this cluster" is imprecise. Which cluster are we looking at here?' Also: were flame cells already described morphologically at this stage, or is this the first description of the protonephridial system for this stage of the life cycle?

We have now clarified which cluster we are talking about in the text. The flame cells have been described using TEM before (Pan, 1980).

Stem Cells: also here you refer to cells as 'bridge' which refers to the configuration of the UMAP. While this is likely a biological representation of a different differentiation state, the nomination of this based solely on the UMAP representation should be avoided.

We have rephrased this.

Figure 5B: What is neuron 6? This was Neuron 3 in Figure 1.

Thank you for spotting these mistakes in the labelling, we have corrected them now.

Ln 421:438 - Here you represent a UMAP representation of the cell positions, but state that the clustering disappears. See comment in Public Review.

Modified accordingly, see response in public review.

Ln 472 "Cells in stem E, F, and G in silico clusters might be stressed/damaged/dying cells or cells in transcriptionally transitional states." Is there any evidence supporting either of these conclusions?

We found that 15.56% of the cells in Stem F expressed the neural marker complexin, leading us to consider the possibility that a fraction of these cells may be neural precursors. Stem F also had some cells with a mitochondrial % near the maximum threshold we set, suggesting they could be experiencing some stress. Since we could not identify clear markers for these clusters, their function and a more specific identity, beyond ‘stem’, is not yet known.

That the two stem cell populations contribute to different parts of the next life cycle stage is interesting. The combined analysis suffers from the same issues as the previous analysis in terms of sample distribution; are the 'grey' sporocyst cells also contributing to the stem A/B (kappa) C/D (delta/phi) clusters? This is not possible to tell from the plot as the miracidia may simply be plotted on the top. A different representation of sample contribution to clusters is warranted.

We have made an alternative visualisation here to demonstrate that the miracidia cells are not plotted on top of the sporocyst stem cells. Unfortunately this visual is hampered as there is not a straightforward way to split the panels. In the figure below, the left pane shows the miracidia cells, and the right pane shows the sporocyst cells. Below that, we have included the original figure for comparison. It can be clearly seen that there are three miracidia tegument cells in the sporocyst tegument cluster, and one sporocyst cell in the miracidia stem cells (Stem E), but the miracidia A/B and C/D stem cells are not plotted on top of any sporocyst cells.

**Author response image 6. sa3fig6:** 

Methods: Why is the multiplet rate estimate at >50% for the unsorted sample?

We have added more detail on this: “The estimated doublet rate was calculated based on 10X loading guidelines and adjusted for our sample concentrations”.

**Reviewer #2 (Recommendations For The Authors):**
(1) The manuscript would benefit from a more careful consideration of what was already known based on previous literature, which would help the authors to better put their results in context. For example, previous work suggested that one of the sporocyst stem cell populations (phi) gives rise to tegument and other temporary larval structures; this appears not to be mentioned here. The model in Figure 7 suggests that two of the stem cell populations are gone at day 15 post-infection; the literature shows that those cells can still be detected at this stage (there are just far fewer of them).

We have added the definition of Kappa, Delta and Phi as per Wang et al (2018) in the stem cell results p13 ln 428.

We have amended Figure 7 to include further elements from the Wang et al (2018) paper that show that mother sporocyst stem cells classified as delta and phi are still detectable on day 15 post-infection in mother sporocysts.

We intentionally didn’t put too much emphasis on fitting our data to the model of Wang et al (2018), because (a) it’s a different life cycle stage and (b) the single cell data the model was based on was from 35 stem cells and gathered using a different method, (c) more recent data (Diaz, Attenborough et al. 2024) with 119 stem cells from sporocysts did not recover the same populations of stem cells. We therefore linked our data to previous literature where it was relevant but focused on being led by the data we gathered (>10,000 stem cells).

(2) To add some detail to the public comment about the lack of clarity about sample sizes and biological replicates, and how this leads to questions about the robustness of the results, Figures 4 B and F show the expression pattern for the same parenchyma marker (Smp_318890) in two different samples. The patterns appear quite distinctive. In B, the cell bodies are so clearly labeled that the signal appears oversaturated. In F the cell bodies are barely apparent. Based on the single-cell clustering, it should be possible to distinguish between Parenchyma clusters 1 and 2 based on the levels of this transcript. Careful quantification of signal intensity from multiple samples across multiple experiments might enable the authors to detect such differences.

The reason the expression patterns look different between panels 4Bii and 4F is that in 4Bii we have manually segmented the nuclei of the parenchymal cells in order to count them, whereas in the images in 4F there is no segmentation. We have made this more clear in this legend now, and also in the legends of Figures 2,3, and 5. If there was any signal intensity difference between parenchyma 1 and 2 cells based on expression of the marker gene, Smp_318890, it was not obvious. We carried out 6 experiments for parenchyma markers, multiplexing the pan-parenchyma marker, Smp_318890, with markers for parenchyma 2 but we were unable to distinguish between the two populations.

(3) The authors find that the "somatic" stem cells in miracidia seem to combine attributes of the previously defined delta and phi stem cells from sporocysts. Because the 3 classes of sporocyst stem cells were defined by expression of nanos-2 and fgfrA, using those probes in in-situ experiments could have helped them resolve whether or not the miracidial cells represent precursors that can adopt either fate or if the heterogeneity is already present in miracidia.

In silico expression of the marker genes for the 3 classes of sporocyst stem cells didn’t support those three classes in the miracidia stem cells (See supplementary table 10). We further subclustered the delta/phi cells to see if we could recover separate delta and phi populations but we were unable to do so. We therefore did not pursue in situ experiments of these genes. We instead prioritised cluster-defining genes in the miracidia stem cell populations rather than cluster defining genes in the sporocyst (defined by Wang et al., 2018), but we still explored these in silico. For example, instead of using klf to define Kappa (Wang et al 2018), we used UPPA to validate the Kappa population as it showed similar expression to klf but higher expression levels and was specific to that population. However, like Wang et al 2018, we did use p53, which is a cluster marker of delta and phi in sporocysts, as it showed clear and high expression in our miracidia delta/phi population. We were guided by our data and our knowledge of the literature. More in depth single cell RNAseq is needed from the mother and daughter sporocyst stages to understand the heterogeneity and fates of these stem populations.

(4) Scale bars should be included throughout the figures and the scale should be defined either on the figure or in the legend. Similarly, all the scales used for velocity and expression analysis should be defined.

We have added scale bars to all figures and legends.

The statements “Gene expression has been log-normalised and scaled using Seurat(v. 4.3.0)”, “Gene expression has been normalised (CPM) and log-transformed using scvelo(v. 0.2.4)”, or “Library size was normalised and gene expression values were log-normalised using SAM (v1.0.1) and Scanpy (v1.8.2)” has been added to all figures as appropriate.

(5) The table entitled In situ hybridization probes (Supplementary Table 15) contains no probe sequences, so any interested reader wishing to use these probes would have to design their own. To ensure the reproducibility of the results presented here, the authors should provide the probe sequences they used.

In Supplementary Table 15 we have added the Molecular Instruments Lot number of all the probes used. Anyone wanting to repeat the experiment can order the same probes from the company.

(6) It is unclear how useful the supplemental figures showing the STRING enrichment analyses will be for readers. Unannotated Smp gene identifiers provide no way to help readers digest the information in these hairballs. It would probably be best to replace the Smp names with useful annotations based on their orthologs; if not, these figures could probably be dropped entirely. (Also, the bottom panel of Supplementary Figure 7 has the word "Lorem" embedded on one of the connecting nodes.)

“Lorem” has been removed.

Many of the genes in these analyses do not have short descriptions, therefore we have used Smp gene identifiers in the STRING analysis supplementary figures. These ‘Smp_’ numbers can be used to search WormBase Parasite, where a description can be found and the history of the gene ID traced. This latter function facilitates searching for these genes in the literature and consistency between versions as gene models are updated.

Minor edits(1) Figures 4A-D aren't cited in the text until after 4E-F are. It seems like moving the section on protonephridial cells (line 364) before the section on tegumental cells (line 345) better reflects the order of the figures.

Thank you for flagging this, we have updated the in-text citations of Figure 4.

(2) In-text references to Sarfati et al, 2021 should be to Nanes Sarfati, as listed in the references. Poteaux et al 2023 is cited in the text, but not in the reference list.

Both of these have been fixed.